

# New ozone-nitrogen model shows early senescence onset is the primary cause of ozone-induced reduction in grain quality of wheat

Jo Cook[1], Clare Brewster[2], Felicity Hayes[2], Nathan Booth[1], Sam Bland[3], Pritha Pande[3], Samarthia Thankappan[1], Håkan Pleijel[4], Lisa Emberson[1]

[1]Environment & Geography Dept., University of York, YO10 5DD, UK
[2]UK Centre for Ecology & Hydrology, Environment Centre Wales, Bangor, Wales, UK
[3]Stockholm Environment Institute at York, Environment & Geography Dept., University of York, YO10 5DD, UK
[4]Biological and Environmental Sciences Dept., University of Gothenburg, SE-40530, Sweden

*Correspondence to*: Jo Cook (jo.cook@york.ac.uk)

**Abstract.** Ozone ($O_3$) air pollution is well known to adversely affect both the grain and protein yield of wheat, an important staple crop. This study aims to identify and model the key plant processes influencing the effect of $O_3$ on wheat protein. We modified the $DO_3SE$-Crop model to incorporate nitrogen (N) processes, and parameterised the $O_3$ effect on stem, leaf and grain N using $O_3$ fumigation datasets spanning 3 years and 4 $O_3$ treatments. Our results show the new model captures the $O_3$ effect on grain N concentrations, and anthesis leaf and stem concentration, well. However, the $O_3$ effect on harvest leaf and stem N

is exaggerated. Further, a sensitivity analysis revealed that, irrespective of $O_3$ treatment, early senescence onset was the primary plant process affecting grain N. This modelling study therefore demonstrates the capability of the $DO_3SE$-CropN model to simulate processes by which $O_3$ affects N content, and thereby determines that senescence onset is the main driver of $O_3$ reductions in grain protein yield. The implication of the sensitivity analysis is that breeders should focus their efforts on stay-green cultivars that do not experience a protein penalty when developing $O_3$ tolerant lines, to maintain both wheat yield and

nutritional quality under $O_3$ exposure.

## 1 Introduction

### 1.1 The importance of wheat for nutrition and the threat of $O_3$

The FAO projects that staple cereals will play a critical role in ensuring food security; particularly in Central and West Asia, and North Africa where wheat provides at least 40% and 47% of dietary calories and protein respectively, compared to ~20%

of dietary calories and protein globally (FAO, 2017). There is a large body of experimental evidence suggesting that current ambient ozone ($O_3$) concentrations in key wheat growing locations are causing substantial productivity losses of equal importance to other, more well known, biotic and abiotic stresses (Mills et al., 2018a; Emberson et al., 2009). Globally, wheat production is reduced by ~7%, though in some regions with high ambient $O_3$ concentrations, such as Northern India, the yield loss is much greater (> 15% in Northern India) (Mills et al., 2018a). There is also a growing body of literature showing that

$O_3$ affects the nutritional quality of the wheat grain and reduces the protein yield (Broberg et al., 2015; Piikki et al., 2008;





Yadav et al., 2020, 2019a). Over the next century, increases to global population and economic growth, and changes to climate and land use, will increase emissions of $O_3$ precursors ($NO_x$ and VOCs) (Fowler et al., 2008). While $O_3$ concentrations are expected to decrease in areas such as Europe, North East America and Japan, concentrations are expected to increase over most of Asia and Africa due to weak regulation of precursor emissions (Fowler et al., 2008). Given Sustainable Development

Goal 2's focus on access to nutritious food, understanding $O_3$ impact on the supply and nutritional quality of wheat grown in regions where $O_3$ concentrations are high/ predicted to increase is crucial (FAO et al., 2020). This will help us understand and address the threats posed by $O_3$ pollution on the ability of future wheat production to meet increasing demand and nutrient requirements (Shiferaw et al., 2013; Mills et al., 2018b).

### 1.2 The impact of $O_3$ on wheat quality and the mechanisms by which damage occurs

Reactive oxygen species (ROS), formed from $O_3$ entering the leaves through the stomata, trigger a series of reactions that culminate in reductions to grain yield and quality (Emberson et al., 2018; Broberg et al., 2015). The nutrient yield of the grains in grams per metre squared is reduced, but the concentration of nutrients (grams nutrient per gram dry matter) in the grains generally increases, as the uptake and remobilisation of nutrients is affected to a lesser extent by $O_3$ than the $O_3$ induced reduction in grain dry matter (DM) (Wang and Frei, 2011; Broberg et al., 2015). Broberg et al. (2015) further found that an

increased grain protein concentration caused the baking properties, quantified by the Zeleny value, Hagberg falling number and dry and wet gluten content, to be positively affected by $O_3$. In some wheat studies, where $O_3$ concentrations are very high, grain protein concentration is decreased, potentially as a result of nitrogen (N) being used for antioxidant production and defence against $O_3$ (Baqasi et al., 2018; Yadav et al., 2020, 2019b; Mishra, Rai and Agrawal, 2013; Fatima et al., 2018).

The main mechanism by which $O_3$ reduces wheat yields and impacts on quality is through accelerated senescence (Emberson

et al., 2018). Wheat cultivars with delayed senescence, stay-green cultivars, have previously been trialled for their potential to offset the reduction in yield that occurs under stressors such as heat and drought stress (Kamal et al., 2019). However, accelerated senescence typically reduces remobilisation of proteins and reduces wheat quality. (Havé et al., 2017; Sultana et al., 2021). It is important to understand the mechanisms by which $O_3$ damages crop yield and influences crop quality for breeding of $O_3$ tolerant cultivars. Section 2.2.1 provides more detail and a mechanistic description of how wheat yields, and

protein, are affected by $O_3$.

### 1.3 The current status of crop modelling with regards to N and $O_3$

Current understanding of the effect of $O_3$ pollution on wheat nutritional quality has been inferred from experimental studies (Mills et al., 2011; Feng, Kobayashi and Ainsworth, 2008; Broberg et al., 2015, 2021). A drawback of experimental work is that it is time consuming and costly, and it can be difficult to control all variables involved. Crop models use environmental

inputs to simulate crop growth, for a range of conditions and stressors, in far less time, using fewer resources than would be required for experimental investigation (Chenu et al., 2017). Developing crop models with experimental data, which provides insights into plant growth processes, allows investigation of realistic plant responses to individual and multiple stressors.



It is possible to simulate the $O_3$ effect on grain protein through incorporating N processes into an existing crop model considering $O_3$ damage, and using a simple conversion factor, such as that from Mariotti, Tomé and Mirand (2008), to convert N to protein. Many models consider N dynamics in wheat (e.g. APSIM-NWheat and CERES-Wheat) (see supplementary Table S1), and others have incorporated $O_3$ damage (e.g. LINTULLC2, WOFOST, APSIM and $DO_3SE$-Crop) (Nguyen et al., 2024; Xu et al., 2023). Some models, such as APSIM, possess the capacity to simulate both $O_3$ effects on yield and grain N, which could, in principle, be used to calculate grain protein and hence provide a measure of grain quality variation under $O_3$ exposure. However, to our knowledge, no simulations have yet been performed on the $O_3$ effect on grain N. Further, these models do not yet include the mechanisms that relate $O_3$ with grain N. No model currently exists that includes the capacity to simulate the reduced remobilisation of N under $O_3$ exposure from the stem and leaf to the grain, an important determiner of wheat protein under $O_3$ exposure (Broberg et al., 2021, 2017; Brewster, Fenner and Hayes, 2024; Brewster, 2023; Chang-Espino et al., 2021).

### 1.4 Objectives

This study aims to develop and use the $DO_3SE$-Crop model to investigate the impact of $O_3$ on wheat grain N content through the following objectives:

1) Identifying the key mechanisms necessary to model N in crops and the influence of $O_3$ on these mechanisms.
2) Developing a N module that can be incorporated into the existing $O_3$ deposition and crop growth model, $DO_3SE$-Crop, to simulate how grain N (and hence protein), is affected by $O_3$ exposure ($DO_3SE$-CropN).
3) Using the developed $DO_3SE$-CropN model to perform a sensitivity analysis to determine which of the $O_3$ damage mechanisms (senescence onset, senescence rate/end, and remobilisation of N) affects grain quality the most.

## 2 Model development

### 2.1 Overview of the DO₃SE-Crop model

The $DO_3SE$-Crop model is used to estimate $O_3$ deposition to a plant canopy and the impacts (biomass and yield loss) caused by stomatal $O_3$ uptake (Emberson et al., 2018). The crop phenology is estimated based on thermal time sums. Photosynthesis is simulated at the leaf level, based on a modified version of the biochemical Farquhar model (Farquhar, Caemmerer and Berry, 1980), and scaled to the canopy level by splitting the canopy into equally sized layers of cumulative leaf area index (LAI). The photosynthetic products from each layer are summed to give the net primary productivity (NPP). The NPP is allocated to the root, stem, leaf or grain based on the plant's developmental stage using the approach of Osborne et al. (2015). $O_3$ transfer from the atmosphere to the leaf is estimated by a resistance scheme incorporating aerodynamic, boundary layer and surface resistances above and within the canopy (Pande et al., 2024). The instantaneous impact of stomatal $O_3$ flux on photosynthesis and the impacts of accumulated $O_3$ flux on senescence, once the cumulative flux exceeds a cultivar specific threshold, are estimated based on the approach of Ewert and Porter (2000) and modified by Pande et al. (2024). Further details of the $DO_3SE$-





Crop model along with a mathematical description can be found in Pande et al. (2024). In this study version 4.39.11 of the DO$_3$SE-Crop model was used (citation to repository to be added before publication).

## 2.2 Development of the N module

### 2.2.1 Identification of which N processes to model

The key plant processes influenced by N and O$_3$ were identified to aid with decisions on which processes to include in the N module for DO$_3$SE-Crop (Figure 1). Figure 1 provides an overview of which processes are included already in the DO$_3$SE-Crop model, which processes will be included in the new N module, and which processes will be excluded. Figure 1 is separated into numbered sections which are explained in the subsequent text:

#1

Reactive oxygen species (ROS), that are formed when O$_3$ diffuses through leaf stomata, trigger a series of physiological and stress responses in the plant that lead to accelerated senescence and a reduced photosynthetic rate (Emberson et al., 2018). ROS delay the response of the stomata to external stimuli, which reduces stomatal conductance (Dai et al., 2019; Paoletti and Grulke, 2010). ROS are detoxified by apoplastic anti-oxidants, but an excess of ROS may overwhelm the anti-oxidant response causing damage to the cell plasma membrane (Emberson et al., 2018; Fatima et al., 2019).

#2

ROS destroy photosynthetic pigments (Emberson et al., 2018). The degradation of photosynthetic pigments by ROS accelerates crop senescence, during which Rubisco, comprising 50% of soluble leaf protein, is broken down to release N for remobilization to other parts of the plant (Feller and Fischer, 1994; Emberson et al., 2018).

#3

The degradation of Rubisco by ROS leads to reduced carboxylation efficiency (Emberson et al., 2018). Together with a reduced electron transport efficiency, photosynthetic rate is reduced (Emberson et al., 2018; Rai and Agrawal, 2012). There is an increase in antioxidant and defence proteins triggered by elevated O$_3$ (Sarkar et al., 2010; Cho et al., 2011; Fatima et al., 2018).

#4

Accelerated senescence as a result of O$_3$ exposure can reduce the green leaf area available for photosynthetic reactions, further decreasing carbon assimilation (Emberson et al., 2018). As a result of accelerated senescence leading to diminished photosynthesis, less photosynthate is produced (Emberson et al., 2018). A larger proportion of available photosynthate is used in respiration and for anti-oxidant production to target ROS (Emberson et al., 2018; Khanna-Chopra, 2012). Under O$_3$ stress, allocation of assimilate to flowers and seeds is prioritised in annual crops such as wheat, reducing the availability for leaves, stems and roots (Emberson et al., 2018).

#5

N taken up by the plant is used to produce all proteins (Lawlor, 2002). Root biomass, and subsequently nutrient uptake, is reduced under stress conditions as assimilate allocation to repair aboveground O$_3$ damage is prioritised over export to the roots





125 (Emberson et al., 2018; Pandey et al., 2018). While $O_3$ can induce senescence and reduce photosynthesis, a higher leaf N can delay the onset of senescence and increase the photosynthetic rate (Pilbeam, 2010; Nehe et al., 2020; Martre et al., 2006; Brewster, Fenner and Hayes, 2024). On the other hand, N deficiency can damage the structure and function of the chloroplasts which could exacerbate $O_3$ impacts on senescence and photosynthetic rate (Kang et al., 2023). Brewster, Fenner and Hayes (2024) found an increase in residual leaf and stem N occurs, potentially as a result of $O_3$ toxicity.

130 #6

Wheat yields are reduced due to the reduced photosynthesis and reduced duration of grain filling (Emberson et al., 2018; Broberg et al., 2015). Wheat grain N comprises N taken up post-anthesis, and N remobilised from the leaves and stem when senescence begins (Havé et al., 2017; Gaju et al., 2014; Nehe et al., 2020; Barraclough, Lopez-Bellido and Hawkesford, 2014). Hence, any damage mechanism which affects grain filling duration, influences the final N content of the wheat grains.

135 Additionally, as Rubisco is a key source of N for grains, once senescence begins, reductions to Rubisco will impact the amount of N that is available to grains (Feller and Fischer, 1994). Brewster, Fenner and Hayes (2024) and Chang-Espino et al. (2021) find evidence of an additional, unknown process, separate to accelerated senescence, that reduces the remobilisation of N under $O_3$ exposure.

Generally, grain protein concentrations are increased under elevated $O_3$, resulting from a relatively smaller decrease in uptake

140 and re-translocation of N relative to the $O_3$ induced decrease in grain dry matter (Wang and Frei, 2011; Broberg et al., 2015, 2019; Triboi and Triboi-Blondel, 2002; Piikki et al., 2008). However, per metre square of crop the starch and grain protein yield is reduced (Broberg et al., 2015; Feng, Kobayashi and Ainsworth, 2008; Gelang et al., 2000). Some wheat cultivars have shown a decrease in grain protein concentration under $O_3$ exposure (Yadav et al., 2019a; Baqasi et al., 2018; Mishra, Rai and Agrawal, 2013). This could be because leaf proteins are being converted to enzymatic antioxidants to provide defence against

145 $O_3$ induced damages, resulting in less proteins available for translocation to the grains (Yadav et al., 2019b; Sarkar et al., 2010; Fatima et al., 2018).







**Figure 1: A flow chart of the mechanisms by which O₃ causes damage to both grain yields and grain N (or protein) in wheat. The different colours and line styles outlining the individual boxes represent whether the process is included in the DO₃SE-Crop model already (blue, dashed outline), not included in the DO₃SE-Crop model (black, solid outline), or included in the N module developed for DO3SE-Crop in this study (green dashed outline, rounded edge boxes). The black connector lines represent interactions between the O₃ damage processes and the green dashed connector lines represent where N processes interact with these. The thinner lines represent interactions that are not included in the DO₃SE-Crop model or the new N module, whereas thicker lines represent the interactions that are included. The figure has been divided into 6 numbered sections for which the mechanisms are described individually in Sect. 2.2.1.**

### 2.2.2 Assessment of existing crop models that include N

Supplementary Table S1 summarises and discusses the similarities between models that simulate plant N dynamics. Most models simulated leaf and stem N by fulfilling the required N demanded by the respective parts from the N uptake pool, with N demand based on a defined minimum and maximum for that organ. The maximum and minimum N concentrations can be set as constants or defined using the phenological stage of the plant, which in turn is based on the accumulation of thermal time or a temperature sum based on a scheme by van Keulen and Seligman (1987). Most crop models define a labile pool of N available to be translocated to the grain and consisting of N available from post-anthesis uptake, non-structural stem N, and N released from leaf senescence. The N released from leaf senescence is calculated in proportion to the decrease in carbon of green leaf area, as N remobilisation is proportional to carbon remobilisation (Havé et al., 2017). Most of the crop models simulated grain N by calculating and fulfilling a N demand, or simulating a rate at which the grains fill with N.

### 2.2.3 Modifications to the existing DO₃SE-Crop model for this study

Prior applications of the DO₃SE-Crop model have assumed that the last 33% of the mature leaf lifespan is when leaf senescence occurs (e.g. Pande et al., 2024). In some of these applications, multiple leaf populations were considered. Given the limitations of available data in parameterising the model for multiple leaf populations, only one leaf population is considered in the present study. As a result, the fraction of mature leaf lifespan that is senescence needed to be modified to instead simulate the fraction of canopy mature leaf lifespan that is senescence. Recent work by Brewster et al. (2024) has shown that the 4th leaves can begin to senesce even before anthesis. Given the importance of senescence in determining N remobilisation (Nehe et al., 2020; Gaju et al., 2014), work by Brewster et al. (2024) was used to re-parameterise the onset of rapid phase senescence as the last 75% of the canopy level mature leaf lifespan for the Skyfall cultivar in DO₃SE-Crop.

### 2.2.4 The DO₃SE-Crop N module for wheat

Based on Sect. 2.2.1 and 2.2.2, we identified key processes for inclusion in the N module as: soil N uptake, partitioning of N uptake between leaf and stem, remobilisation of N in leaf and stem to the grains, grain filling with N, and O₃ effects on grain N. At the present stage of the modelling, we do not include any processes relating to usage of N for antioxidant production or utilisation of photosynthate for above ground repair due to lack of data for parameterisation. Full details of equations, sources of equations and model parameterisations are available in Appendix A. Briefly:

(a) Soil N uptake





Pre-anthesis, daily N uptake from the soil is proportional to the increase in LAI and stem mass that day, along with any N deficit that has accumulated over the plant's life, following the work of Soltani and Sinclair (2012). Post-anthesis,

we use the formulation from SiriusQuality (Martre et al., 2006) which links post-anthesis N uptake with the capacity of the stem to store N. Pre- and post-anthesis we define a maximum N uptake which cannot be exceeded. Since we did not have data to calibrate for the effects of N stress, the present model assumes optimal soil N availability.

(b)   N partitioning

Pre-anthesis, N uptake is allocated to the leaf and stem in accordance with the increase in LAI or stem mass that day,

as commonly used by other crop modellers (Sect. 2.2.2). The specific equations used closely follow those of Soltani and Sinclair (2012).

(c)   N remobilisation

After anthesis, N remobilisation from the stem to the grains begins. N is released from senescing leaves in accordance with the decrease in LAI that day. Released N is stored in the stem where it is available to the grain. The combination

of N released from leaf senescence and non-structural stem N creates the labile pool of N for grain filling.

(d)   Grain N

The N in the labile pool can be transferred to the grain, or it can remain as part of the stem. In contrast to other crop models, the proportion of labile N transferred to the grain each day follows a sigmoid function. The sigmoid was chosen as it uses only two extra parameters ($\alpha_N$ and $\beta_N$) which allows the start and rate of grain fill with N to be customised

without the addition of much complexity. The fraction leaving the labile pool increases as the plant develops.

$$N_{to\_grain} = N_{labile} * \frac{1}{1 + exp(-\alpha_N(DVI - \beta_N))} \tag{1}$$

where $N_{to\_grain}$ represents the amount of N leaving the labile pool ($N_{labile}$) to the grains, $DVI$ represents the development index of the plant in DO₃SE-Crop (Pande et al., 2024). The $N_{to\_grain}$ profile for different parameterisations of $\alpha_N$ and $\beta_N$ is shown in Appendix A, Fig. A3.

(e)   Direct effect of O₃ on grain N

The fraction of N remaining in the leaf and straw increases with O₃ exposure (Broberg et al., 2017, 2021). Additionally, in the work of Brewster et al. (2024) the same effect is observed, where a lower proportion of N stored in these parts at anthesis is moved to the grains. Little data is available on this effect, so we used all available existing data from Broberg et al. (2017) and Brewster et al. (2024) to produce a linear regression of the % of N remaining in the leaf and stem at harvest as a function of M12 (the common metric for the two studies defined as the 12 hour mean O₃ concentration

during daylight hours (Guarin et al., 2019)). The results of this can be seen in Fig. 2.





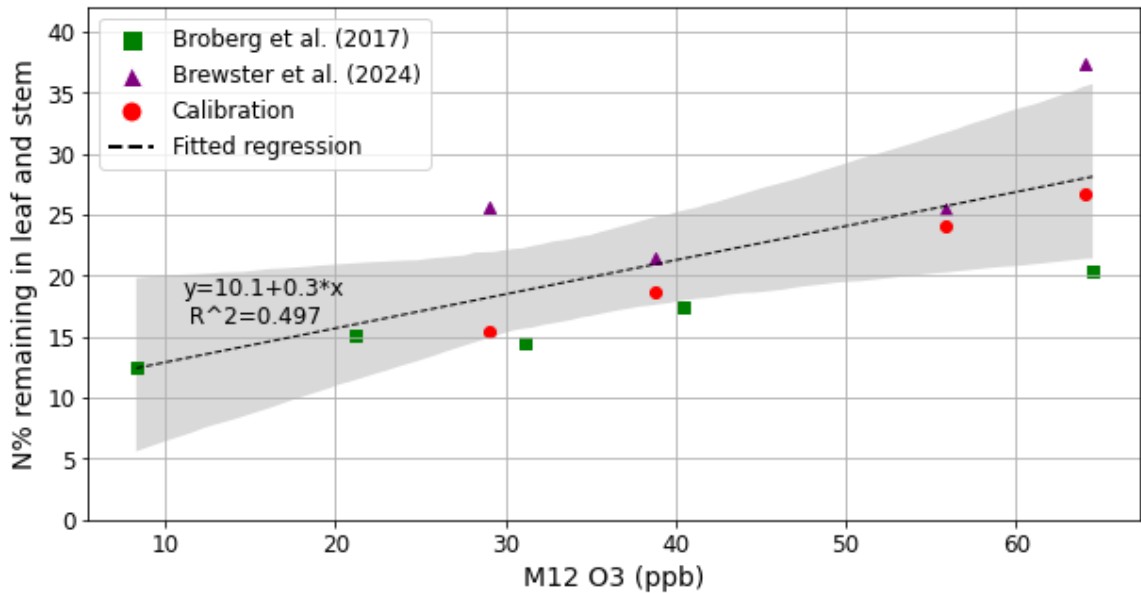

**Figure 2: The % of N remaining in the leaf and stem as a function of M12 for studies by Broberg et al. (2017) (green square markers) and supplementary data obtained from Brewster, Fenner and Hayes (2024) (purple triangular markers). The grey area represents the 95% confidence interval of the fitted regression (dashed linear line) and the R² of the regression is given on the figure. Overlaid are red, circular markers showing the effect of the calibrated $m_{leaf}, c_{leaf}, m_{stem}$ and $c_{stem}$ on overall remobilisation.**

The minimum allowed leaf and stem N concentrations were varied to optimise the grain N% and harvest leaf and stem N% simulations, whilst making sure the % of N remaining in the leaf and stem was within the 95% CI of the remobilisation regression. The form of the regressions representing the minimum leaf and stem N concentrations under O₃ exposure are given in Eq. (2) and (3) respectively.

$$\frac{[N_{leaf,min}]}{1 \ gN \ LAI^{-1}} * 100 = m_{leaf} * \frac{[O_{3,M12}]}{1 \ ppb} + c_{leaf} \tag{2}$$

$$\frac{[N_{stem,min}]}{1 \ gN \ DM^{-1}} * 100 = m_{stem} * \frac{[O_{3,M12}]}{1 \ ppb} + c_{stem} \tag{3}$$

where $[N_{leaf,min}]$ is the minimum leaf N concentration in gN per unit of LAI, $[N_{stem,min}]$ is the minimum stem N concentration in gN per g of stem DM, and $[O_{3,M12}]$ is the concentration of O₃ using the M12 metric in ppb. The parameterisation for Eq. (2) and (3) is given in Table 1. Further details of the process by which the best parameters were obtained is given in Sect. 4 of Appendix A.





**Table 1: the calibrated values for the newly developed regressions describing how the minimum leaf and stem N concentrations vary under differing O₃ concentrations.**

| Parameter | Description | Unit | Value |
|---|---|---|---|
| $m_{leaf}$ | Gradient of Eq. (2) | / | 0.798 |
| $c_{leaf}$ | Intercept of Eq. (2) | / | 10.89 |
| $m_{stem}$ | Gradient of Eq. (3) | / | 0.0138 |
| $c_{stem}$ | Intercept of Eq. (3) | / | 0.2293 |

(f)     Indirect effect of O3 on grain N

In the DO₃SE-Crop model, O₃ accelerates the onset and rate of senescence (Pande et al., 2024). In this N module, remobilisation of N from senescing leaves occurs once senescence begins. Further, no N is remobilised from the leaves once senescence is complete. DO₃SE-Crop also simulates the impact of O₃ on the rate of photosynthesis and, consequently, biomass production and leaf area expansion (Pande et al., 2024). In this N module, leaf area determines accumulation of leaf N and stem biomass determines accumulation of stem N, providing an indirect link between O₃

damage in the DO₃SE-Crop model and the newly developed N module.

In combination, the N processes and DO₃SE-Crop processes are integrated to form the DO₃SE-CropN model as shown in Figure 3. Simply, N uptake is partitioned in accordance with demand from the leaf and stem. The N available to the grain comes from senesced leaf area, post anthesis N uptake and non-structural stem N. The amount of N that is transferred to the grain from this available pool is calculated using a sigmoid function. The fraction of N that is available to the grain from the

leaf and stem is modified in accordance with daily O₃ concentrations. Further details of the equations used, and processes involved are given in Appendix A. In this study version 1.0 of the N module was used, and the corresponding code can be found at (citation to repository to be added before publication).



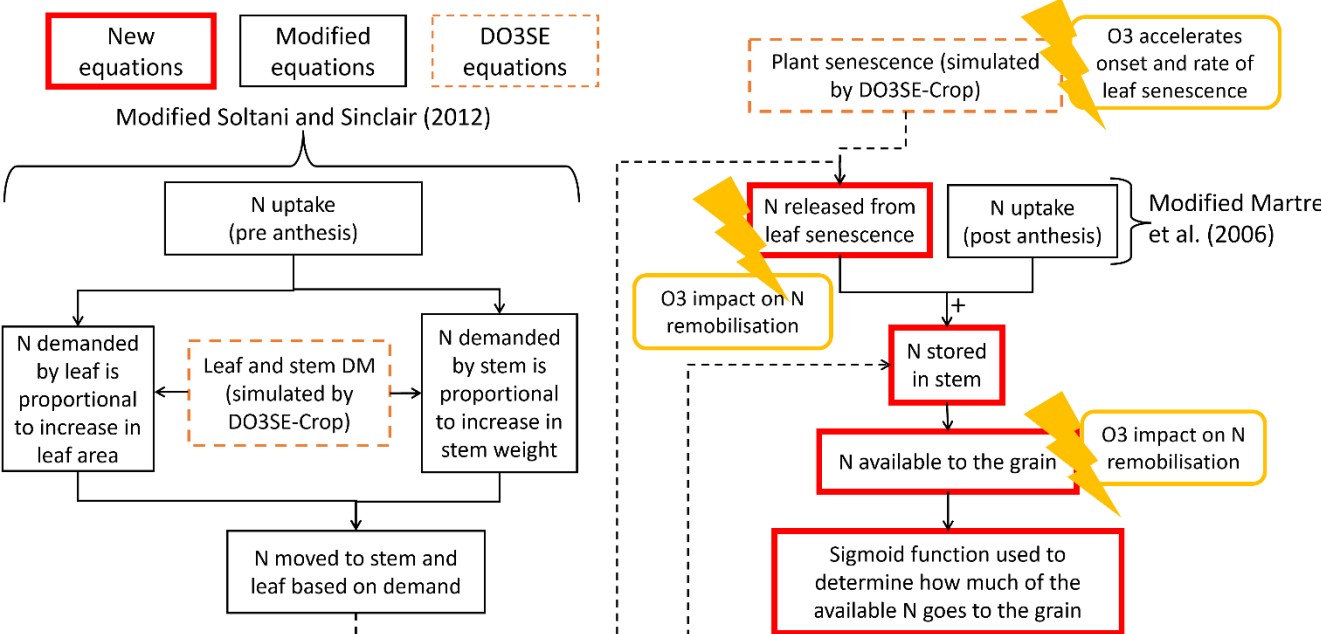

**Figure 3: Simplified overview of the DO₃SE-Crop N module. Red outlined boxes show the processes developed in this study. Orange outlined boxes are indicating where the N module takes outputs from the DO₃SE-Crop model. Black outlined boxes indicate where equations were taken from the existing literature and modified for the current study.**

## 3 Parameterisation and calibration of DO₃SE-Crop and new N module

Experimental data have been collated and gap filled (using the AgMIP-Ozone gap-filling methodology (Emberson et al., 2021)) to calibrate and evaluate the DO₃SE-Crop model and newly developed N module. Further details of the gap filling methods can be found in the Supplementary materials.

### 3.1 Experimental datasets

Data from the Centre for Ecology and Hydrology, Bangor, from the years 2015, 2016 and 2021 were used to calibrate the N module for DO₃SE-Crop. For each year, the Skyfall wheat cultivar was grown in solar domes under 4 O₃ treatments, with median O₃ concentrations ranging from 29 – 61.1 ppb. Across the 3 years, the wheat was planted between February 23[rd] and April 15[th] and harvested between August 11[th] – August 17[th]. O₃ fumigation occurred between stem-elongation (GS30) and harvest (mid-August) (Brewster, Fenner and Hayes, 2024; Broberg et al., 2023; Osborne et al., 2019 with supplementary information from the authors). In this study, only treatments where the plants were well watered and experienced no N or temperature stress were used. No grain data was used for the year 2021 as the plants did not put on any grain (see Brewster, Fenner and Hayes (2024) for further detail). A tabulation of data available from each year can be found in supplementary Table S2. Further details of the experimental set up for all years can be found in Brewster et al. (2024), Broberg et al. (2023) and



Osborne et al. (2019) and supplementary data obtained from the authors was used for model development (citation to data repository to be added before publication).

**3.2 Model calibration and evaluation**

Model calibration was performed in stages, with the calibrated parameters indicated in brackets. Definitions of DO$_3$SE-Crop
parameters, along with the value they were calibrated to and the units are given in supplementary Table S3.

1) Phenology ($T_b, T_o, T_m, TT_{emr}, TT_{flag,emr}, TT_{astart}, TT_{amid}, TT_{harv}$)

2) Photosynthesis and respiration ($V_{cmax,25}, kN, J_{max,25,}, m, R_{dcoeff}, R_g$)

3) DM allocation and yield, and O$_3$ effect on yield and senescence ($\alpha_{root}, \beta_{root}, \alpha_{leaf}, \beta_{leaf}, \alpha_{stem}, \beta_{stem}, \Omega, \tau,$ $E_g, \gamma3, \gamma4, \gamma5, \text{CLsO3}$)

4) N allocation and O$_3$ effect on N remobilisation ($m_{leaf}, c_{leaf}, m_{stem}, c_{stem}$ (Sect. 2.2.4, Table 1))

This sequential calibration prevents later adjustments caused by the interdependencies between parameters at different calibration stages. It was necessary to calibrate O$_3$ effects on yield at the same time as the DM allocation and yield parameters, as O$_3$ still influences yield, even in the low O$_3$ treatments. $V_{cmax,25}$ and $J_{max,25,}$ had been measured and so were fixed to their experimentally measured values to limit the numbers of parameters to calibrate. Further, data on photosynthesis at different
light concentrations allowed the determination of the rate of dark respiration, and hence we fixed the parameter controlling dark respiration in DO$_3$SE-Crop. A combination of genetic algorithm and manual calibration was then used to calibrate chosen parameters to achieve the desired output variable. The genetic algorithm is not always the most suitable for model calibration, as it can give parameterisations that maximise the R$^2$, but don't make sense physiologically. In cases where unrealistic parameterisations were given, a manual "by-eye" calibration process was also used. By varying one parameter at a time to
understand it's effect on a desired output, realistic parameterisations were chosen. For all calibrations the R$^2$ and RMSE were used to assess the fit between observed data and simulations. Generally, 50% of the combined data for all years, for the low and very high O$_3$ treatments were used in the calibration, with the remaining data used in the evaluation along with 100% of the medium and high O$_3$ treatment data for all years. A tabulation of the parameters calibrated for, and the values they were calibrated to, are given in supplementary Table S3 along with further details of the calibration process. Parameterisations
relating to the newly developed N module are discussed and presented along with their equations in Appendix A.

The model was evaluated by calculating the RMSE and R$^2$ of the linear regression between observed and simulated values of phenology dates, grain DM, stem and leaf N and grain N%. More emphasis is placed on the relative O$_3$ impact on yield and quality than simulations of absolute values, as the aim of the study was to develop a model that can capture relative O$_3$ impacts on crop quality.





### 3.3 Sensitivity analysis

Sensitivity analyses are used to determine the proportion of variance in a desired model output attributed to a variation in the model input (Saltelli et al., 2008). In this study, we use a sensitivity analysis to identify and rank the sensitivity of grain N to different plant processes simulated by DO$_3$SE-Crop and the new N module. We identified 3 key mechanisms that can influence grain quality in the crop model: senescence onset, the end/ duration of senescence and the O$_3$ interruption of N remobilisation of the leaf and stem. A preliminary sensitivity analysis was conducted to identify the key parameters in DO$_3$SE-Crop influencing these processes. After reduction, 4 parameters which contribute the greatest to the variance of output variables representing these processes were identified for the sensitivity analysis (see Table 2). We use an extended Fourier amplitude sensitivity analysis (eFAST) to explain the variation in a chosen output variable attributed to varying selected input variables over a given range (Saltelli et al., 2008). The eFAST method is a commonly used method for sensitivity analyses and has previously been used by crop modellers to improve calibration (Silvestro et al., 2017; Vazquez-Cruz et al., 2014). The benefit of eFAST over other forms of sensitivity analysis is that it allows the interactions between model parameters to be quantified, it can sample the entire parameter space, and it is robust for non-linear relationships (Saltelli et al., 2008; Cariboni et al., 2007). These benefits make it a useful tool for complex systems such as crop models, where interacting, non-linear processes are common (Saltelli et al., 2008; Cariboni et al., 2007). The Python library SaLIB was used for all sensitivity analyses (Herman and Usher, 2019). The first sensitivity index, S1, quantifies the uncertainty in the output variable that is attributed to varying only that parameter. The total sensitivity index, ST, quantifies the uncertainty in the output variable that is attributed to varying a chosen parameter in combination with the other selected parameters (Saltelli et al., 2008). The range of values for the sensitivity analysis were taken from the theoretical maximum and minimum in the DO$_3$SE-Crop model for those mathematical equations. The ranges for $\gamma 4$ and $\gamma 5$ were determined using the breakpoint method, as described by (Pande et al., in review). For the leaf and stem remobilisation equations, the minimum gradient is 0 as this assumes no O$_3$ effect on remobilisation for that plant part, and the maximum gradient was calculated by assuming that the other plant part has had as close to zero O$_3$ impact on remobilisation as is mathematically possible in the equation formulation.



**Table 2: The parameters included in the sensitivity analysis along with a specification of the values for the ranges that they were**
**varied in during the analysis**

| Parameter | Unit | Explanation | Minimum | Maximum |
|---|---|---|---|---|
| $\gamma 4$ | - | $O_3$ long-term damage co-efficient determining onset of senescence | 0.1 | 10 |
| $\gamma 5$ | - | $O_3$ long-term damage co-efficient determining maturity | 0.1 | 1.5 |
| $m_{leaf}$ | - | Gradient of regression determining minimum leaf N concentration under $O_3$ exposure (influences how much leaf N is available for remobilisation) | 0 | 3.024 |
| $m_{stem}$ | - | Gradient of regression determining minimum stem N concentration under $O_3$ exposure (influences how much stem N is available for remobilisation) | 0 | 0.0312 |

## 4 Results

### 4.1 End of season grain DM and N% in grain, leaf and stem

Figure 4 shows the evaluation of the grain DM, and grain, leaf and stem N% simulations. Leaf and stem N data were only available in 2021, hence the leaf and stem plots only use data from 2021. Additionally, for 2021, the plant did not put on any
grain (the reason for which is unknown (Brewster, Fenner and Hayes, 2024)) which meant it was not possible to use the grain DM or grain N data for that year. However, since this was the only year of data for which stem and leaf N% measurements were available and the plants developed and flowered normally, the decision was made to proceed with these data for stem and leaf parameterisation. For 2016, the model captured the grain DM and the grain N% more precisely than for the year 2015. In 2015, the under-estimate of grain DM resulted in an overestimation of grain N%. The stem and leaf N% at anthesis is better
simulated than at harvest. However, in both the stem and the leaf, N concentrations are over-estimated at both measured growth stages, despite the calibration showing the remobilisation of N from the leaf and stem under the differing $O_3$ concentrations was well captured (see Figure 2). The $R^2$ values (calculated using scikit Learn, developed by Pedregosa et al. (2011)) for grain



DM and grain N% are negative, implying that the model simulations are worse than using the mean of the observed data (scikit-learn developers, 2023).



**Figure 4: Output of the DO₃SE-Crop grain dry matter simulations at harvest (a), and the newly developed N module simulations of grain N% at harvest (b), leaf N% at anthesis and harvest (c), and stem N% at anthesis and harvest (d). The simulations shown are for the evaluation datasets only. The evaluation data available contained grain dry matter and N% for 2015 and 2016, whereas for 2021 the plant did not put on any grain, so this data was not used for evaluation. Leaf and stem N% data was only available for 2021. The RMSE and R$^2$ of the observed versus simulated data (not including error bars) are shown on the plots. The error bars represent the maximum and minimum of the experimental data, excluding outliers, for comparison with simulations. The leaf N% figure contains data for both the flag and 2$^{nd}$ leaf. DO₃SE-Crop and the new N module do not discriminate between these so simulations of leaf N% for flag and 2$^{nd}$ leaf are the same**

The relative yield (RY) loss of the 2015 simulations is much better simulated than the 2016 simulations (Figure 5). However, the R$^2$ is 0, meaning that the simulations work equally well as when using the mean of the observed data (scikit-learn developers, 2023). When considering grain quality, the increase in grain N% that occurs as O₃ concentrations increase, is simulated very effectively with an R$^2$ of 0.6 and a RMSE of 4.9%. The % decrease in grain N content (grain N content measured in grams of N per m$^2$ of crop) is not simulated as well as the change in grain N concentration, as is seen from the lower R$^2$





(0.3) and greater RMSE (14.4). Further, Figure 5c shows that the model had trouble capturing the large differences in grain N

content that occurred in 2016, compared to the much smaller differences in 2015.

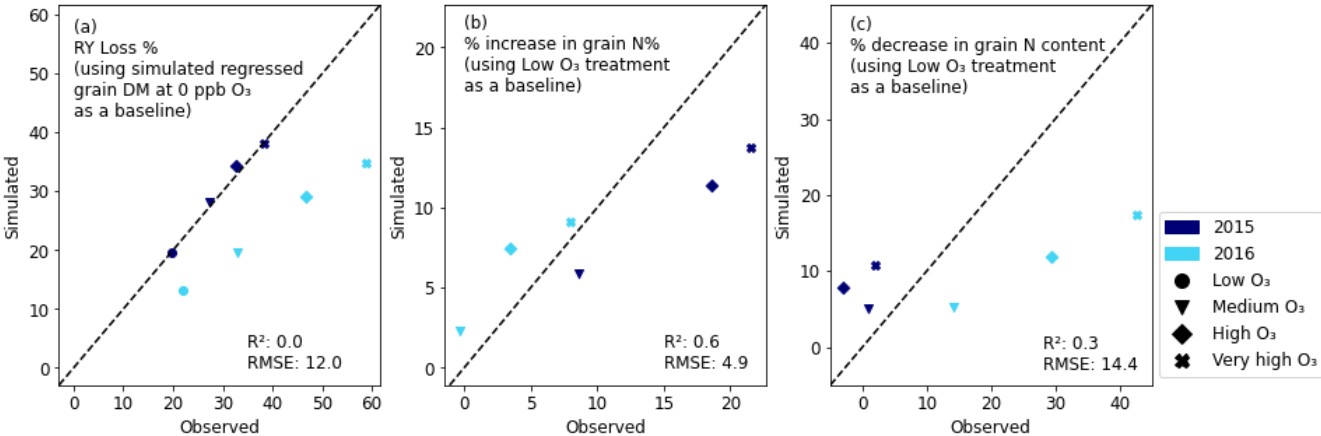

**Figure 5: Relative plots of the evaluation data. (a) the relative yield (RY) loss of the grain DM when using the grain DM at 0 ppb (obtained by regressing the simulated and observed yields) as a baseline. (b) the % increase in grain N% and (c) the % decrease in grain N content (gm$^{-2}$) when using the low O₃ treatment as a baseline. The RMSE and R$^2$ of the observed versus simulated data (not**

**including error bars) are shown on the plots.**

### 4.2 Seasonal profiles of grain DM, grain N content and N% in the grain, leaf and stem

The profile of grain DM accumulation over time is shown in Figure 6. From the profiles we see that initially accumulation of

grain DM in both years is slow, then at days 200 and 192 for 2015 and 2016 respectively, there is a rapid increase. From the

plots we can see the O₃ effect on grain DM accumulation begins around 5-10 days post-anthesis.

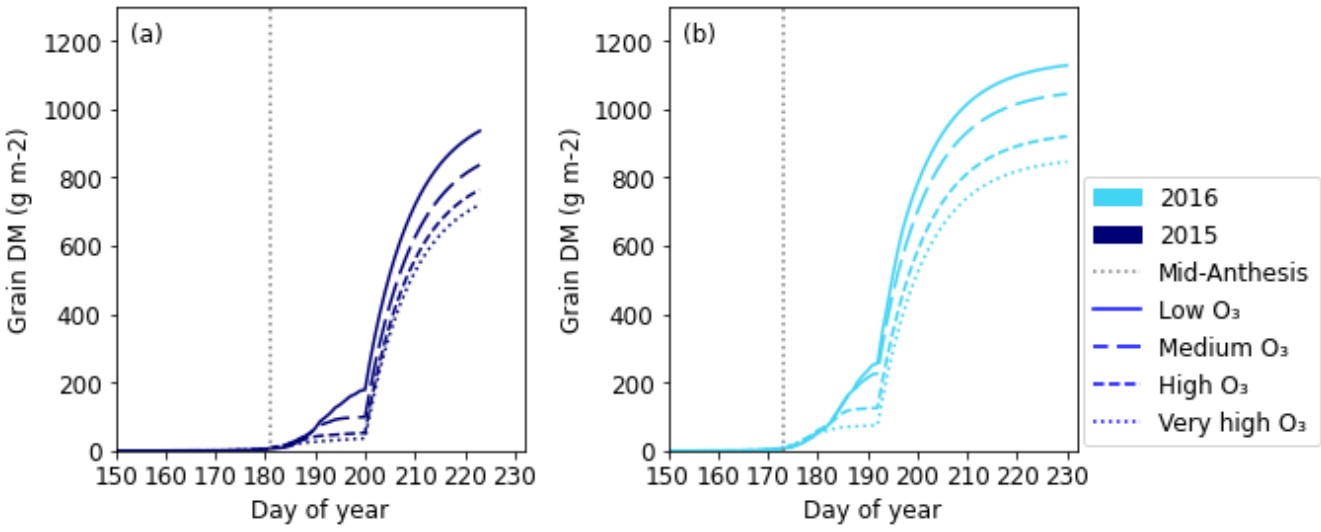


**Figure 6: The profile of simulated grain DM for 2015 (a) and 2016 (b). Mid-anthesis is indicated on the graph as a vertical dotted line and the different line styles on the plot represent the simulations for the different O₃ concentrations.**





Figure 7 shows the change in simulated grain N% and grain N content as a function of time under the different $O_3$ concentrations. As $O_3$ concentrations increase, the absolute grain N content in $gm^{-2}$ (Figure 7c and Figure 7d) decreases for
both years. Figure 7a and Figure 7b appear to show a very sharp increase in grain N% as the grain starts filling with N after anthesis, and then after approximatley 20 days N concentration starts to decrease. This rapid increase is due to a difference in the accumulation rates of grain DM and N in the model, and is not representative of a plant process (see Supplementary Fig. S1). Due to the large spike in initial N concentrations, it is difficult to see the effect of $O_3$ on the end N concentrations. Therefore, the end profiles of the grain N concentrations were enlarged in Figure 7a* and Figure 7b*. Once magnified, it is
possible to see the increase in grain N% with increasing $O_3$ concentrations.

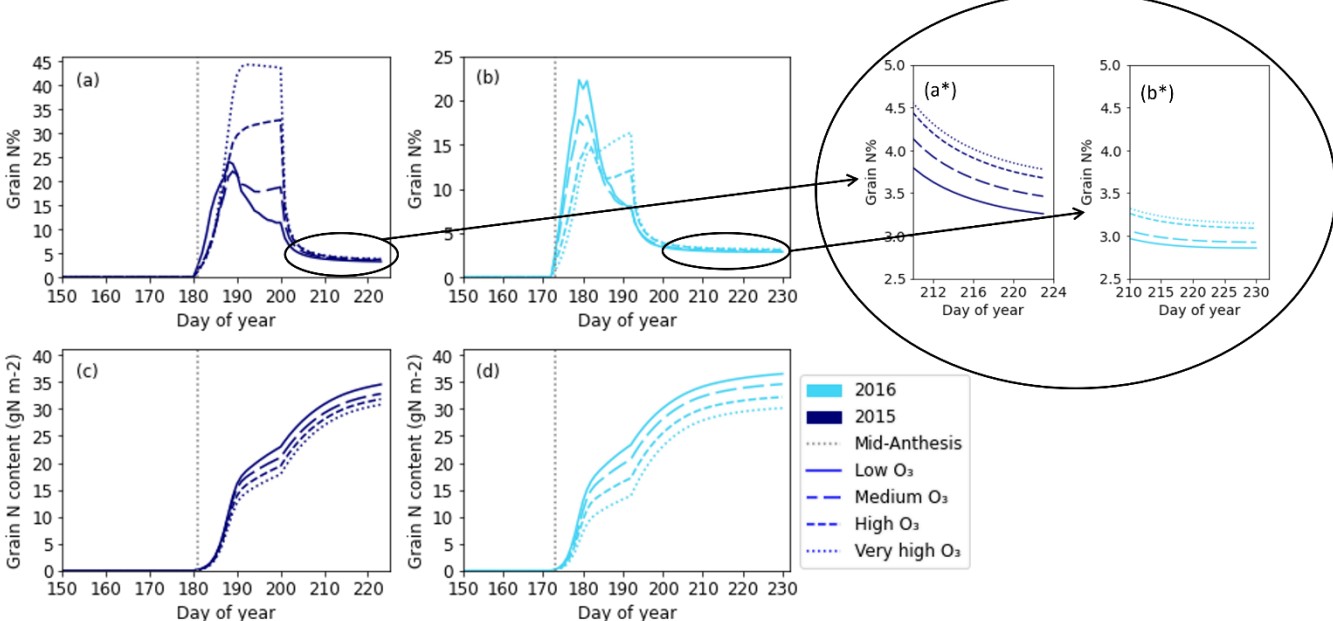

**Figure 7: The simulated grain N% for 2015 (a), and 2016 (b), and the simulated grain N content in grams per metre squared for 2015 (c) and 2016 (d). The different line styles represent the different $O_3$ concentrations. Mid anthesis is indicated on the graph as a dashed vertical line for each year. The end points of figures (a) and (b) have been enlarged and are represented as figures (a*) and**
**(b*) respectively so that differences at the end points can be distinguished.**

Figure 8 shows the seasonal profile of leaf and stem N content and % under differing $O_3$ concentrations. Simulations of leaf and stem N% (Figure 8a and 8b) are relatively constant at their target N concentration (see Appendix A for target N explanation), until anthesis, since the model assumes no limitations to soil N uptake. The leaf and stem N content increase in line with increasing biomass. Post-anthesis, the stem begins to transfer N to the grain, and so the N concentration and content
in the stem decreases (Figure 8b and Figure 8d). The remobilisation of N from the stem to the grain continues provided the stem N concentration does not decrease below the minimum (Figure 8b and Figure 8d). The levelling off of the stem N% in Figure 8b shows the minimum stem N concentration for that $O_3$ treatment has been reached. At higher $O_3$ concentrations the stem remobilises less N to the grains and the final concentration of N in the stem is greater (Figure 8b and Figure 8d). Initially leaf N% and content decreases faster in the simulated wheat plants experiencing greater $O_3$ concentrations, as senescence





begins earlier in these treatments (Figure 8a and Figure 8c). Then, at around 20 days after mid-anthesis the O₃ effect on

remobilisation takes over, and the leaf N% and content are greater under increased O₃ concentrations, due to the O₃ inhibition

of N remobilisation.

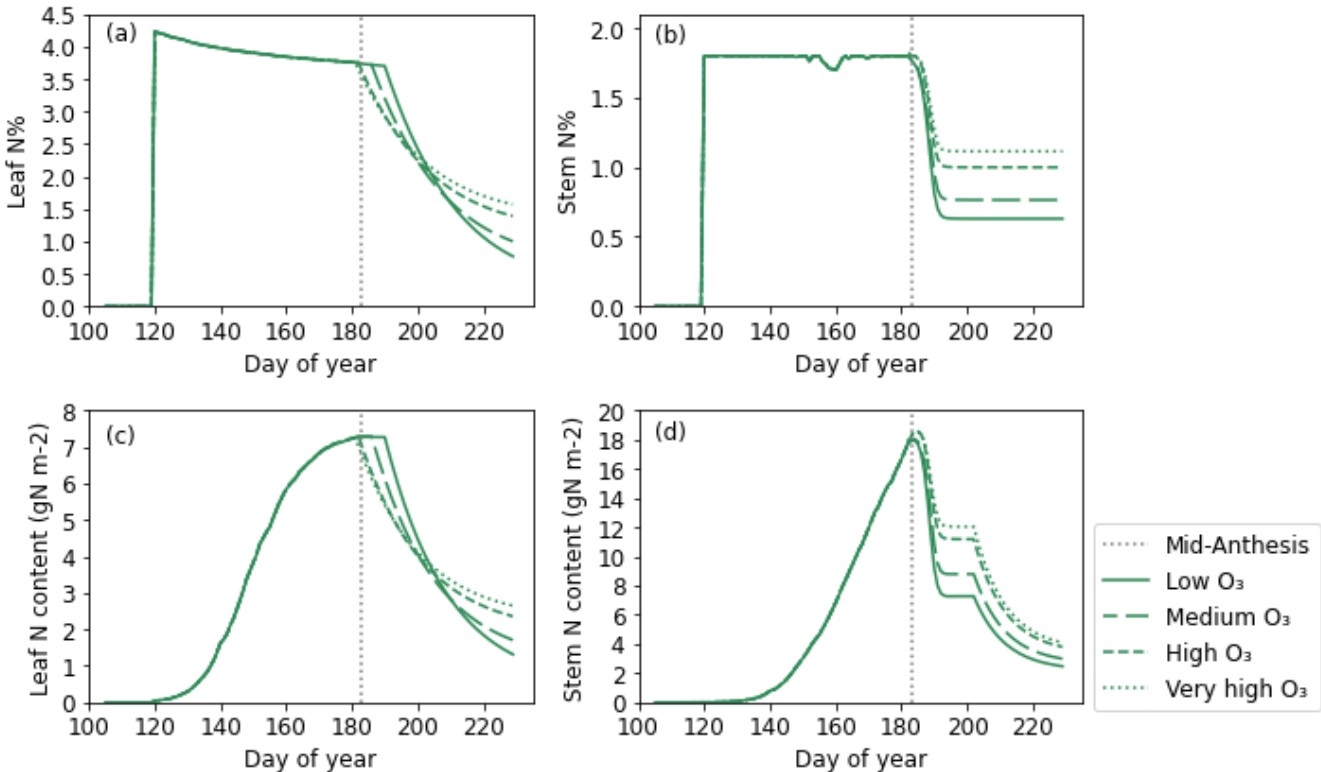

**Figure 8: The simulated leaf (a) and stem (b) N%'s along with the simulated absolute N content in grams per metre squared for the**
**leaf (c) and stem (d). Mid-anthesis is indicated on the graph as a vertical dotted line and the different line styles on the plot represent**
**the simulations for the different O₃ concentrations. These plots are for the 2021 simulations only.**

## 4.3 Sensitivity analysis results

Figure 9a and Figure 9b illustrate the results of the sensitivity analysis study, and show that greater than 60% of the variance

in both grain N content and grain N% in simulations of all O₃ treatments is explained by varying the parameter controlling the

onset of leaf senescence. Absolute grain N content (gm⁻²) is more sensitive to variations in senescence onset than grain N%,

and grain N% is more sensitive to variations in senescence-end than grain N content. A threshold of ST > 0.1 was used by

Silvestro et al. (2017) to identify influential parameters in their sensitivity analysis. In this study, senescence onset and

senescence end were found to be the only influential parameters on grain N content and grain N%. The effect of varying the

leaf and stem remobilisation accounts for less than 2% of the variance in both grain N% and grain N content for all O₃

simulations and can be considered non-influential. The interactions between the parameters were close to zero for the grain

N%, as shown by a small difference between S1 and ST. Whereas a stronger interaction was seen when considering absolute





grain N as the output. The negligible ST terms for leaf and stem remobilisation imply that the larger ST observed for senescence onset in Figure 9a must be due to the interaction between senescence onset and end.

**Figure 9: Results of the sensitivity analysis. Lighter bars represent the first sensitivity index, S1, and the darker bars represent the total sensitivity index, ST. The different colours represent the different $O_3$ treatments and for clarity are also indicated on the x ticks. Braces group together the results of the sensitivity analysis for a particular plant process. The single model parameter chosen to represent each plant process in this analysis is described in Sect. 3.3. This graph shows the averaged results for the 3 different years. The coloured bars represent the mean value of the sensitivity index for all 3 years, while the error bars represent the maximum and minimum values for that sensitivity index achieved in the runs for the 3 years. Figure (a) shows the results of the sensitivity**



**analysis when considering the absolute grain N content in gN m$^{-2}$ as the output parameter. Figure (b) shows the results of the sensitivity analysis when considering the percentage of N in the grain as the output parameter (100*gN gDM$^{-1}$)**

## 5 Discussion

### 5.1 Evaluation of grain DM simulations

#### 5.1.1 Grain DM simulations at harvest

The relative yield (RY) loss of the wheat in 2015 was more accurately simulated than in 2016, despite the grain DM being better simulated in 2016 than 2015 (Figure 4a and Figure 5a). The variability of whether grain DM or RY loss is better simulated occurs because it is not possible to calibrate grain DM independently of the O$_3$ effect on yield loss, due to overnight O$_3$ concentrations of >20 ppb in the solar domes, which give a pollutant effect on yield even in the lowest O$_3$ treatment.

Subsequently, there is a trade-off between achieving a greater accuracy in either grain DM or RY loss simulations as the current model construct was not able to capture both at the same time. In this study, the decision was made to give priority to greater accuracy on the relative O$_3$ effect on yield, so that we could better test our simulations of the relative effect of O$_3$ on grain quality. Further, the splitting of data for calibration and evaluation was randomised. The randomisation resulted in the calibration simulations for 2015 having a slightly lower grain DM, and hence the grain DM for 2015 in the evaluation

simulations was underestimated.

#### 5.1.2 Seasonal profile of grain DM accumulation

The profiles of grain DM accumulation follow an approximate sigmoid shape, with a "bump" in the grain DM profile occurring when the stem begins to allocate DM to the grain (Figure 6a and 6b). Approximately 5-10 days after mid-anthesis we see a reduction in grain DM accumulation for the higher O$_3$ concentrations. While profiles of grain DM accumulation have not yet

been studied for O$_3$, they have been studied for other stressors such as drought. Given that both drought stress and O$_3$ damage are ROS mediated we can expect their effects on the seasonal profiles of DM and N to be similar, where the stresses occur continuously throughout the growing season (Khanna-Chopra, 2012; Emberson et al., 2018). Liu et al. (2020) observed an increase in grain weight for drought-stressed wheat (as compared with the well-watered wheat) 7 days after anthesis, when the first post-anthesis measurements were made. By 28 days post-anthesis the grain weight of the well-watered treatment surpassed

that of the water stressed wheat until the final measurement 42 days post-anthesis (Liu et al., 2020). Similarly, Zhang et al. (2015) observed the grain weight for drought stressed wheat was greater than the well-watered treatment 12-20 days after anthesis, however, by 36 days post-anthesis the grain weight of the well-watered wheat was greater than the drought stressed wheat. In maize, a decrease in the kernel fresh weight under drought stress was visible after 10 days, with the first post-anthesis observations occurring 5 days after anthesis (Guo et al., 2021). The authors found that the drought stressed maize always had

a lower kernel fresh weight, until the final measurement 30 days post-anthesis (Guo et al., 2021).



In two of the studies which considered the dynamic profiles of individual plant grain DM accumulation under drought stress as compared with a well-watered treatment, the profiles did not show a decrease at every measured timepoint, as we observed in Figure 6. The accumulation of DM or starch under drought stress surpassed that of the well-watered/irrigated treatment before being overtaken again closer to harvest (Liu et al., 2020; Zhang et al., 2015). However, those profiles were based on

individual grains in Liu et al.'s (2020) study, and the thousand grain weight of the 4 main spikes in Zhang et al.'s (2015) study. Neither of those values take into consideration the reduced number of grains per plant that occurs under drought stress (Mariem et al., 2021). Reduced grain numbers are also observed under $O_3$ stress (Broberg et al., 2015). Therefore, a consistently lower grain DM profile when presented as a grain DM per metre squared (as in this study) may be expected for increasing levels of stress.

Together, these data suggest that differences in grain DM accumulation under drought stress are evident as early as 7-10 days post-anthesis. Therefore, it is not unreasonable that our simulations to show a decrease in grain DM accumulation in $O_3$ exposed wheat 5-10 days after mid-anthesis. Further, considering the grain DM per metre squared, we may expect to see a consistently lower grain DM for plants experiencing greater $O_3$ stress.

## 5.2 Evaluation of grain N simulations

**5.2.1 Grain N simulations at harvest**

Harvest grain N% is simulated well for 2016, when the grain DM is more closely captured (Figure 4a and 4b). However, in 2015, when the grain DM is under-estimated, the grain N% is over-estimated. The over-estimation of grain N% is reduced when the observed, rather than simulated, grain DM values are used to calculate grain N% (data not shown). This suggests that the newly developed N module can simulate the absolute grain N% under differing $O_3$ treatments accurately, provided

grain DM is simulated well.

There is a large difference in how well the model captures the relative increase in grain N% under elevated $O_3$ concentrations, compared with the relative decrease in grain N content (gN m$^{-2}$) (Figure 5b and 5c). The model was calibrated to grain N% as the response of grain N% to increased $O_3$ concentrations was more consistent between the two years than the decrease in N content. The observed decrease in N content was highly variable between the two years as there were large differences in the

grain N content of the wheat at harvest. The reason for the large difference is unclear, though it could partially be due to the differences in grain DM between the two years. Since the grain N content was so different between the 2015 and 2016 datasets for all treatments, the model struggled to match both years when simulating the decrease in grain N content. Additionally, there is an interdependence in the model between changes in grain DM, grain N content and grain N%. Changing grain DM or grain N content subsequently changes grain N%, making the calibration process more difficult, which is an additional reason

why it was difficult to calibrate for both grain N% and grain N content (see supplementary Fig. S1). If the wheat in 2021 had put on grain, the grain N measurements would have been invaluable in determining which of 2015 and 2016 had the more



common response of grain N to $O_3$ for the Skyfall cultivar. Having at least 3 datasets for model calibration allows outliers to be more easily identified, and more physiologically representative data selected for calibration.

### 5.2.2 Seasonal profile of grain N accumulation

The seasonal grain N content profiles (Figure 7c and 7d) match well with profiles seen in experimental work (Nagarajan et al., 1999; Bertheloot et al., 2008) and that can be constructed from available wheat N data (see supplementary Fig. S2). There is a "bump" in our simulated grain N content (Figure 7c and 7d) which matches that of our simulated grain DM profile (Figure 6a and 6b). The "bump" occurs because as stem DM decreases (due to remobilisation), it increases the available N for the grain. It is not thought that this "bump" represents the actual rate by which the grain fills with N, but instead occurs due to the

complex interconnections between leaf senescence, leaf N remobilisation, stem N remobilisation and required N for the grains in the model.

The seasonal profiles of grain N content (Figure 7c and 7d) show that the $O_3$ effect begins to be distinguishable at ~10 days after mid-anthesis for both years, with the model simulating lower rates of accumulation of grain N content when wheat is exposed to higher concentrations of $O_3$. To our knowledge, no studies exist that describe how the seasonal profile of grain N

content accumulation varies under differing $O_3$ concentrations. However, again using the fact that both $O_3$ and water stress are ROS mediated (Khanna-Chopra, 2012; Emberson et al., 2018), the response of grain N accumulation under water stress as compared to a well-watered treatment could be comparable to grain N accumulation under different $O_3$ concentrations. When Nagarajan et al. (1999) exposed four wheat cultivars to water deficits post-anthesis, the shape of the grain N content response was very different between all 4 cultivars. For 2 cultivars, the profile of grain N content accumulation was very similar between

the stress and control groups. Whereas, for the other 2 cultivars, water stress had a substantial negative impact on the total amount of N accumulated in the grain and the rate of grain N content accumulation as compared to the control treatment (Nagarajan et al., 1999). In a study measuring N accumulation with and without irrigation, Panozzo and Eagles (1999) found that the irrigated wheat accumulated N more slowly per grain, but continued accumulating for longer and ultimately had a higher N content per grain. In the current study, our simulations show that the grains accumulate less N content under $O_3$,

which occurs due to the explicit modelling of a reduced remobilisation rate under higher $O_3$ concentrations (Figure 7c and Figure 7d and Eq. (2) and (3)).

Seasonal profiles of grain N% (Figure 7a and 7b) under different $O_3$ treatments have a less consistent pattern than the equivalent grain N content profiles. Initially, there is a very rapid increase in grain N% which then levels off and decreases. Nagarajan et al. (1999) measured grain N and grain carbon (C), both in mg per plant, of 4 wheat cultivars at 3 time points post-anthesis for

a water stress and a control treatment. We used these data to construct time profiles of grain N% (see supplementary materials and Fig. S4). The constructed profile of grain N% over time generally shows an initial increase which tends to decrease or level-off. Only one of the cultivars under both water-stress and the control treatments showed a grain N% profile that consistently increased (Fig. S4). Further, Panozzo and Eagles (1999) measured grain weight in mg, and mg N per grain for 7 time points post-anthesis, again allowing a profile of grain N% under dry and irrigated treatments to be constructed. The



profiles constructed from Panozzo and Eagles (1999) show that the wheat grain N% decreases from 2.5% and levels off to 2% approximately 21 days after anthesis, matching the shape of our grain N% profile, though our peak grain N% is far too large at 15-45%. In our study, the reason for the large peak in grain N% is that N accumulation occurs more rapidly after anthesis than grain DM, leading to a greater concentration of grain N (see supplementary Fig. S1). If the initial rate of grain fill with N was slower, or grain DM accumulation was faster, our grain N% profiles would likely match the shape and magnitude of those

constructed from Panozzo and Eagles (1999) or Nagarajan et al. (1999). It would be helpful to have measurements of grain N content and DM for multiple time points after anthesis under varying $O_3$ treatments, to develop a temporal understanding of grain N% response to $O_3$ for model parameterisation.

In 2015, grain N% is already higher in the elevated $O_3$ treatments around 10 days after anthesis, whereas in 2016 grain N% is initially higher in the lower $O_3$ treatments and then around 20 days after mid-anthesis, the elevated $O_3$ show a higher grain N%

(Figure 7a and 7b). The difference in response of grain N% is due to the differences in the simulated rates of grain N and grain DM accumulation between the years. In the grain N% profiles constructed from Panozzo and Eagles (1999) (see supplementary Fig. S3), the irrigated (non-stressed) wheat had a lower final grain N% than the dry (stressed) treatment and the difference was clear to see from > 10 days post-anthesis. Toward harvest, the effect of increased $O_3$ concentrations on increasing the grain N% is seen approximately 20 days after anthesis, when the initial sharp increase levels off and the concentrations reach more

reasonable values.

## 5.3 Evaluation of stem and leaf N simulations

### 5.3.1 Stem and leaf N simulations at anthesis and harvest

The anthesis leaf and stem N% are captured well by the model but the effect of $O_3$ on harvest leaf and stem N% is exaggerated (Figure 4c and d). Although the differences between harvest leaf and stem N% were non-significant (Brewster, Fenner and

Hayes, 2024), there appears to be a slight decrease in final leaf and stem N% under the medium $O_3$ treatment and a subsequent increase for the high and very high $O_3$ concentrations. A potential reason for the decrease under medium $O_3$ concentrations could be an effect called hormesis, where a stressor initially induces a greater productivity in the plant and then, past a given threshold, has a negative effect (Agathokleous, Kitao and Calabrese, 2019). While it could be argued that a hormesis effect is present for the data of Brewster, Fenner and Hayes (2024) in Figure 2 of this study, there are only 4 $O_3$ treatments which means

it is not possible to parameterise the minimum point of the hormetic response. Whereas Broberg et al. (2017) had 5 $O_3$ treatments, yet did not observe a hormetic response, and instead found it linear. If future experimental work looks at the $O_3$ impact on N remobilisation from the leaf and stem to the grain, it would be helpful to place a greater emphasis on $O_3$ treatments between 30-60 ppb to determine a potential turning point at which higher $O_3$ concentrations start to limit N remobilisation. This would enable parameterisation of a non-linear hormetic response for N remobilisation in wheat under $O_3$ exposure.



### 5.3.2 Seasonal profiles of stem and leaf N accumulation

Our simulations of leaf and stem N content and N% over time (Figure 8) show that they reach a peak before anthesis and decrease after anthesis, which is also shown by the stem and leaf N profiles constructed from available experimental data (see supplementary Fig. S2). The levelling-off of the stem N content (Figure 8d) profile at approximately 190 days is a result of the model construct in that no N was required for the grains from the stem at that point.

Currently, there are no data described in the literature on the effect of $O_3$ on leaf or stem N status in crop plants over the course of the growing season to compare with the profiles produced in this modelling study (Figure 8). However, one study by Bielenberg, Lynch and Pell (2002) did measure the variation in stem and leaf N content, over time, in hybrid poplar exposed to elevated $O_3$. In plants receiving the same N treatment, increased $O_3$ generally reduced the N content of the leaves and stem at each measurement point. Generally, the temporal profiles of N content had the same shape regardless of $O_3$ treatment (Bielenberg, Lynch and Pell, 2002). In our study, the stem N content was greater at every stage post-anthesis for greater $O_3$ concentrations due to the reduced remobilisation of nutrients under $O_3$ exposure (Brewster, Fenner and Hayes, 2024). By contrast, the leaf N content was initially reduced in wheat experiencing greater $O_3$ concentrations due to accelerated senescence, but the effect of the reduced remobilisation eventually outweighed the senescence effect, leading to greater N content in $O_3$ stressed wheat at harvest.

### 5.4 Suggestions for improving the representation of plant growth in DO₃SE-CropN

It has been suggested that the reduced remobilisation of leaf and stem N under $O_3$ exposure occurs as a result of reduced N use efficiency, as $O_3$ accelerated senescence shortens the grain filling period (Broberg et al., 2021, 2017). However, Brewster et al. (2024) found an increase in residual flag leaf N concentration under $O_3$ exposure, despite not finding a difference in senescence onset, suggesting that the accelerated senescence and subsequent reduced N remobilisation efficiency is not the only factor increasing the residual N concentration in plant parts. Some researchers have suggested the increase occurs as defence proteins accumulate (Brewster, Fenner and Hayes, 2024; Sarkar et al., 2010). In the present study, we simulate the accelerated senescence that occurs under $O_3$ exposure, and we model the reduced remobilisation using simple linear equations (Eq. (2) and (3)). Future work to understand the mechanism for the increase in residual N in the stem and leaf would be useful, to modify the model to better represent the existing plant processes.

Brewster, Fenner and Hayes (2024) found that N fertilisation reduced the N left in the leaf and stem at harvest under $O_3$ exposure, delaying senescence and protecting chlorophyll against $O_3$ damage. The current model construct does not simulate this effect and assumes the plant receives optimum N. No N stress or extra fertilisation was considered. Future iterations of this model incorporating soil N processes could use the work of Brewster, Fenner and Hayes (2024) to include N fertilisation and model the ameliorative effect of variable N fertilisation on $O_3$ damage. Additionally, this first iteration of the model does not include any feedback of plant N status on photosynthetic or growth processes. Future research may want to consider the feedbacks between leaf N levels and photosynthetic rate, with higher leaf N increasing photosynthetic rate, which could offset



O<sub>3</sub> induced reductions (Pilbeam, 2010). Researchers may also want to consider the influence of a higher leaf N in potentially delaying $O_3$ induced early senescence onset (Nehe et al., 2020; Martre et al., 2006).

## 5.5 Sensitivity analysis results

The sensitivity analysis (Figure 9a and 9b) showed that the effect of varying leaf and stem remobilisation contributed little, if at all, to variations in grain N content or grain N% under the differing $O_3$ treatments. Further, the most influential parameter on grain N content and grain N%, was the senescence onset; this does not change regardless of the $O_3$ treatment simulated. These results align with existing literature, which show that a shorter duration for grain fill leads to less time for nutrient remobilisation, subsequently impacting grain quality (Havé et al., 2017). Therefore, it is expected that the senescence

parameters would have a large influence on grain N. Further, since the onset of leaf senescence is the beginning of leaf N remobilisation to the grain, it has a larger influence on grain N content and concentration than the end of leaf senescence (Havé et al., 2017). Therefore, for the Skyfall cultivar, under the $O_3$ conditions simulated, senescence onset has greater influence on grain quality than $O_3$ interruptions to remobilisation from an unspecified process. In our sensitivity analysis we observed a difference in the magnitude of S1 (the uncertainty in the output variable that is attributed to varying only that parameter) and

ST (the uncertainty in the output variable that is attributed to varying a chosen parameter in combination with the other selected parameters) (Saltelli et al., 2008) between the different $O_3$ treatments. It is unclear why this effect occurred. It isn't possible to determine whether the magnitude of S1 and ST is anomalous for the low or medium $O_3$ treatments, or whether a pattern exists at all in S1 or ST between $O_3$ treatments since the present study considers data on one cultivar for one location only.

## 5.6 Sensitivity analysis results

The sensitivity analysis results pose interesting questions around the importance of cultivars having an earlier or delayed senescence for maximising grain protein under $O_3$ exposure. Generally, a delayed onset of leaf senescence decreases grain protein content as there is a delay to the start of N remobilisation from the leaves to the grain (Havé et al., 2017; Sultana et al., 2021). A delayed senescence also decreases grain protein%, due to the reduction in grain N content and an increased length of photosynthetic activity that increases grain yield (Sultana et al., 2021; Nehe et al., 2020; Bogard et al., 2011). However, this is

not always the case as there are several interacting effects from the environment (e.g. temperature, drought, $O_3$ or pathogens), N application and gene expression (Zhao et al., 2015; Nehe et al., 2020; Bogard et al., 2011; Gaju et al., 2014; Sultana et al., 2021). Stay-green cultivars have previously been identified as allowing plants to maintain their photosynthetic capacity under heat and drought stress conditions (Kamal et al., 2019). Since $O_3$ damage is ROS mediated, similar to the damage from heat and drought stress (Khanna-Chopra, 2012), it is expected that stay-green cultivars will provide a similar increased yield for $O_3$

exposed wheat by delaying the stress induced early senescence onset. The impact of using stay-green cultivars on wheat grain protein under $O_3$ stress conditions is yet to be investigated. However, a decrease in grain protein content under $O_3$ exposure may be likely due to the delayed onset of remobilisation (Havé et al., 2017; Sultana et al., 2021). Zhao et al. (2015) genetically modified wheat plants to investigate the response of senescence, grain yield and grain N% when a senescence delaying gene





was over-expressed. In the genetically modified wheat, the grain yields were similar to the control, but grain N% was increased
(Zhao et al., 2015). Therefore, stay green wheat cultivars that do not experience a grain protein penalty should be considered
by breeders and investigated for their suitability under differing $O_3$ concentrations.

**5.7 Potential model applications**

Simulations of crop N content can be easily converted into protein through the use of a simple conversion factor, or linear
regressions if N or water stress conditions are present that have not been accounted for previously in the simulation (Mariotti,
Tomé and Mirand, 2008; Liu et al., 2018; Tkachuk, 1969). Therefore, grain protein% can easily be obtained from grain N%.
Further, by using protein%, amino acid concentrations can be simulated using regressions developed by Liu et al. (2019),
allowing the model to be extended to simulate protein quality. In addition, by building on the work of Broberg et al. (2015) it
would also be possible to link changes in N content under $O_3$ exposure, to changes in other grain mineral contents, such as
zinc, magnesium and starch. Such relations would be simple to integrate given the model already simulates $O_3$ effects on N,
and there seem to be similarities between the effect of $O_3$ on N and the effect of $O_3$ on other minerals (Broberg et al., 2015).
These improvements would increase the nutritional relevance of the model.

The DO$_3$SE-Crop model takes inputs of temperature, PPFD, $CO_2$ concentrations and precipitation. There is also a built in soil
moisture module which can simulate the effect of water stress on stomatal $O_3$ flux (Büker et al., 2012). Because of these
features, the DO$_3$SE model is an ideal candidate for simulating the combined effects of $O_3$ pollution and climate change on
crop yields. With the newly developed N module, this would allow the user to determine how $O_3$ pollution and climate change
effects may interact to affect crop yield, quantity and quality, and hence dietary nutrition.

**6 Conclusions**

In summary, this study identified the key mechanisms for modelling N in wheat as soil N uptake, partitioning of N taken up
from the soil between the leaf and stem, remobilisation of N from the leaves and stem to the grain, and the impact of $O_3$ on N
remobilisation. Using these key processes, a new model was developed that can be used in combination with the existing $O_3$
deposition crop model, DO$_3$SE-Crop, to simulate the $O_3$ impact on wheat N. The newly developed model is the first to simulate
the effect of $O_3$ on N in any plant species. After evaluation, a sensitivity analysis was applied to the model to identify the key
plant process that affects grain quality under $O_3$ exposure. It was found that $O_3$ induced early senescence onset was the key
plant processes affecting grain quality under $O_3$ exposure, regardless of $O_3$ treatment. We recommend that breeders focussing
on stay-green cultivars aim to develop cultivars that do not suffer a protein penalty. If such cultivars can maintain their yield
and quality under abiotic stresses, they may also be tolerant to $O_3$ in terms of both yield and crop quality; testing this would
be beneficial to further understand the implications of $O_3$ on global food and nutritional security.





**Appendix A. DO₃SE-CropN**

Detailed below are the equations and references for the N module of the DO₃SE-Crop N module. We found Wang & Engel's (2002) approach to writing up their crop model to be very effective so we take inspiration from their approach and present a description of the processes and references in the text, and tabulate the specific equations, with their corresponding references in each section. The values of parameters used in the module and their corresponding sources are tabulated also. For a full

mathematical description of the phenology, photosynthesis, carbon allocation and ozone damage processes of DO₃SE-Crop, please refer to Pande et al. (2024).

At the current stage of development, the N module is not integrated within DO₃SE-Crop. It requires the output file from a DO₃SE-Crop run in order to perform the N simulations.

**A.1 N Uptake**

Pre-anthesis a maximum N uptake ($NUP_{pre,max}$, in $gN\ m^{-2}\ day^{-1}$) is defined (Soltani & Sinclair, 2012). The actual N uptake by the crop pre-anthesis ($NUP_{pre}$, $gN\ m^{-2}\ day^{-1}$) is calculated using the work of Soltani and Sinclair (2012). Daily N uptake is the sum of the N associated with the increase in LAI that day ($LAI_{growth}$, $m^2\ day^{-1}$), the increase in dry matter of the stem that day ($DW_{stem\_grth}$, $g\ DW\ m^{-2}\ day^{-1}$) and the N deficit that day ($N_{stem,deficit}$, $g\ N\ m^{-2}\ day^{-1}$), that has been accumulated over the plant's life:

$$NUP_{pre} = DW_{stem\_grth} * \left[N_{stem,target}\right] + LAI_{growth} * \left[N_{leaf,target}\right] + N_{stem,deficit}.\qquad (A4)$$

$\left[N_{stem,target}\right]$ is the target N concentration of the stem and $\left[N_{leaf,target}\right]$ is the target N concentration of the leaf (Soltani & Sinclair, 2012). The $N_{stem,deficit}$ is defined as the difference between the target stem N for its mass, and its current N content ($N_{stem}$):

$$N_{stem,deficit} = DW_{stem} * \left[N_{stem,target}\right] - N_{stem}.\qquad (A5)$$

where $DW_{stem}$ is the dry weight of the stem in $g\ m^{-2}$. If $NUP_{pre} > NUP_{pre,max}$ we set $NUP_{pre} = NUP_{pre,max}$, otherwise, the crop takes up N equal to $NUP_{pre}$.

Post-anthesis, N uptake is a function associated with the capacity of the stem to hold nitrogen. The basis for this equation is taken from SiriusQuality (Martre et al., 2006). The potential N uptake post anthesis ($NUP_{post,pot}$ $gN\ m^{-2}\ day^{-1}$), is calculated as follows:

$$NUP_{post,pot} = NUP_{post,max} * \frac{T_{grainfill} - T}{T_{grainfill}}\qquad (A6)$$

where $NUP_{post,max}$ is the maximum N uptake post-anthesis, $T_{grainfill}$ is the thermal time (in °C days) between the end of grain filling (i.e. harvest in the present model) and the start of anthesis, and $T$ is the current thermal time (in °C days) since



anthesis. Subsequently, the stems target capacity to hold N is calculated and compared to the current amount of N stored in the stem. If the current stem N, $N_{stem}$ in $g\ N\ m^{-2}$, exceeds the target capacity, no N is taken up that day (equation (A7)).

$$NUP_{post} = 0\ if\ \ N_{stem} \geq DW_{stem} * [N_{stem,target}]$$
(A7)

If the current stem N is less than the target capacity, the N taken up is equal to the minimum of the stem's current capacity and the potential uptake.

$$NUP_{post} = min(NUP_{post,pot}, DW_{stem} * [N_{stem,target}] - N_{stem})$$
(A8)

**Table A1: Citations and names for equations describing soil N uptake**

| Equation number | Equation name | Developed according to… |
|---|---|---|
| (A1) | Actual plant N uptake pre-anthesis | (Soltani & Sinclair, 2012) |
| (A2) | N stem deficit | (Soltani & Sinclair, 2012) |
| (A3) | Potential daily N uptake post-anthesis | (Martre et al., 2006) |
| (A4) | Post-anthesis N uptake if current stem N is greater than target | developed in this study to ensure stem does not get an unlimited supply of N |
| (A5) | Post-anthesis N uptake if current stem N is less than target | (Martre et al., 2006) and adapted in this study to account for current stem N |


**Table A2: Default and present parameterisations for the N module along with citations as to where the original default value was obtained**

| Parameter | Original Value | Value used in this study | Source for original value |
|---|---|---|---|
| $NUP_{post,max}$ | $0.4\ g\ N\ m^{-2}\ day^{-1}$ | $0.4\ g\ N\ m^{-2}\ day^{-1}$ | (Martre et al., 2006) |
| $NUP_{pre,max}$ | $0.25\ g\ N\ m^{-2}\ day^{-1}$ | $0.65\ g\ N\ m^{-2}\ day^{-1}$ | (Soltani & Sinclair, 2012) |
| $[N_{stem,target}]$ | $0.015\ g\ N\ g^{-1}\ DW$ | $0.018\ g\ N\ g^{-1}\ DW$ | (Soltani & Sinclair, 2012) |
| $[N_{leaf,target}]$ | $1.5\ g\ N\ m^{-2}$ | $1.65\ g\ N\ m^{-2}$ | (Soltani & Sinclair, 2012) |




## A.2 N partitioning to the stem and leaf

Pre-anthesis, the equations describing how the uptake of N is split between the leaf and stem are based on the work of Soltani and Sinclair (2012), and no N is allocated to the grains at this time. The stem and leaf have a defined target and minimum N concentration which can be calibrated for different wheat cultivars. In Soltani and Sinclair (2012) both the target and minimum stem and leaf N concentrations are constants. In this study, the minimum N concentrations of the leaf and stem are variable based on the ozone concentrations. This allows a reduction in remobilisation of N from the leaf and stem to the grain, as found

by Brewster et al. (2024), under higher ozone concentrations. It is easiest to see the structure of the N partitioning code in Figure A1. The write up of pre-anthesis N partitioning to the leaves and stem is split into 2 sections based on whether the stem is experiencing a N deficit (i.e. stem N is less than its minimum).

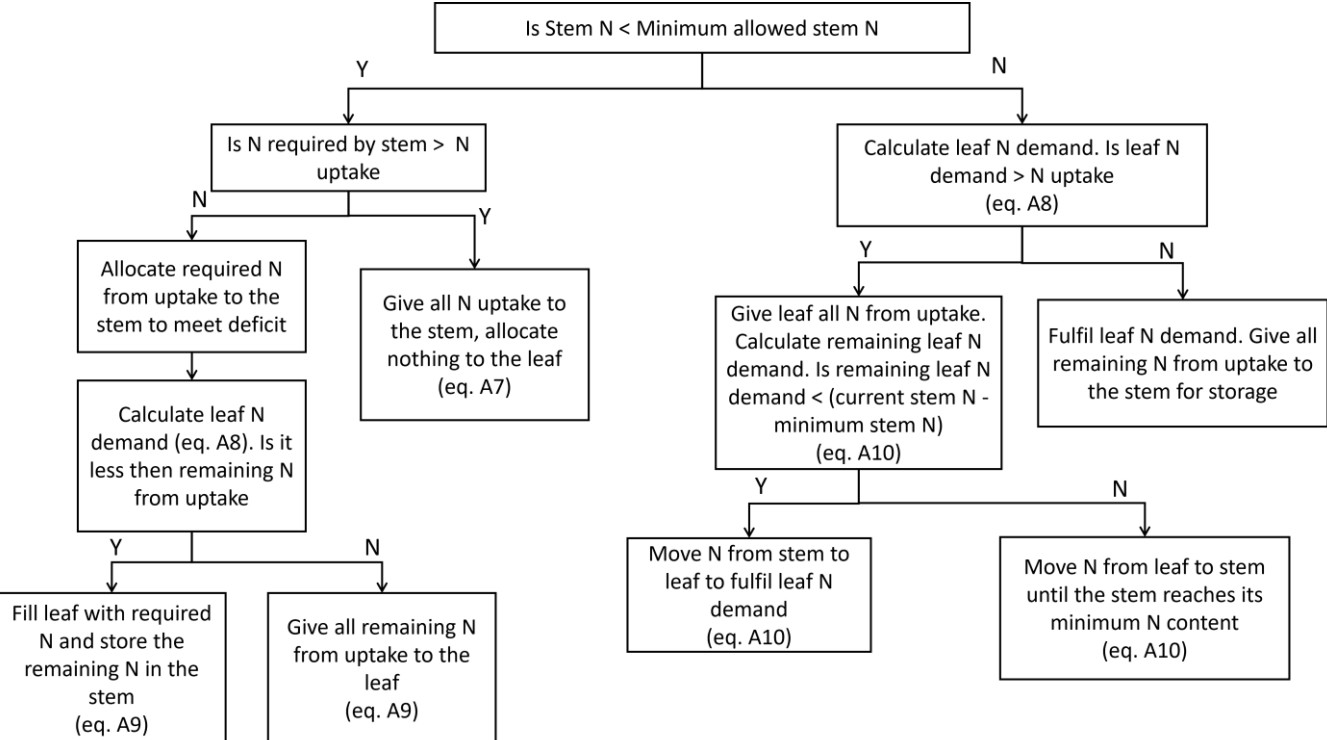

**Figure A1: Figure showing the allocation of N to the stem and leaf before anthesis. Y and N represent "yes" and "no" respectively.**
**Equation definitions are referenced appropriately. Where equation numbers are not indicated additional information can be found in the text. Equations and model structure for N allocation are based on those of Soltani & Sinclair (2012) and modified for the purposes of this study.**

Firstly, the current stem N concentration is compared to the minimum allowed stem N concentration, $\left[N_{stem,min}\right]$ .

### A.2.1 If Current stem N concentration is less than the minimum stem N concentration (N deficit)

If the current stem N concentration is less than the minimum concentration, then allocation of N to the stem is prioritised.





$$N_{req,stem} = DW_{stem} * [N_{stem,min}] - N_{stem} \tag{A9}$$

where $N_{req,stem}$ is the N required by the stem to meet its deficit in $g\ N\ m^{-2}$. If $N_{req,stem}$ is greater than the amount of N taken up that day ($NUP$, $g\ N\ m^{-2}$), the N entering the stem pool ($N_{into,stem}$, $g\ N\ m^{-2}$) is capped at the amount of N taken up.

$$if\ N_{req,stem} > NUP_{pre} \begin{cases} N_{into,stem} = NUP \\ N_{leaving,leaf} = 0 \\ N_{leaving,stem} = 0 \\ N_{into,leaf} = 0 \end{cases} \tag{A10}$$

Equation (A10) has been modified from Soltani & Sinclair (2012) who allow leaf area to senesce and provide the stem with N
if there is not enough to meet the stem's N demand. The senescing of leaf area to provide N was removed in the current iteration of DO₃SE-CropN. Currently, there is no interdependencies between the N module and DO₃SE-Crop. The DO₃SE-Crop model runs and then the N module is applied to the output of DO₃SE-Crop to calculate crop N. It is therefore not possible to senesce leaf area to provide N in the current version of the N module as the leaf area was already determined in DO₃SE-Crop.

If there was enough N to meet the deficit of the stem, then $N_{into,stem} = N_{req,stem}$ and $N_{leaving,stem} = 0$. Then, the N required for maintaining leaf area growth at its target N concentration ($N_{req,leaf}$, $g\ N\ m^{-2}$) is calculated using equation (A11).

$$N_{req,leaf} = LAI_{growth} * [N_{leaf,target}] \tag{A11}$$

If $N_{req,leaf}$ can be fulfilled by the N left over after maintaining the stem growth at minimum N concentration, then N is partitioned to the leaves and any leftover N is partitioned to the stem for storage. If the leftover N cannot fulfil the demand of the leaf, then the remaining N from uptake is partitioned to the leaves (Soltani & Sinclair, 2012).

$$N_{into,leaf} = \begin{cases} N_{req,leaf} & if\ N_{req,leaf} \leq NUP - N_{into,stem} \\ NUP - N_{into,stem} & if\ N_{req,leaf} > NUP - N_{into,stem} \end{cases} \tag{A12}$$

**A.2.2 If Current stem N concentration is not less than the minimum stem N concentration (no N deficit)**

If the stem N is not below its minimum, then N is first allocated to the leaves. The N required by the leaves is calculated in accordance with equation (A11). If the N required by the leaves is less than the N uptake that day, the N required by the leaves is transferred to them: $N_{into,leaf} = N_{req,leaf}$ and $N_{leaving,leaf} = 0$. Subsequently, the remaining N from uptake is transferred to the stem, $N_{into,stem} = NUP - N_{into,leaf}$ and $N_{leaving,stem} = 0$

If the leaves required more N than was taken up by the crop, the extra demand is fulfilled by using some of the stem N. The N from the stem is removed only until the stem reaches its minimum concentration.

$$N_{leaving,stem} = \begin{cases} 0 & if\ N_{req,leaf} \leq NUP \\ \min(N_{req,leaf} - NUP,\ N_{stem} - (DW_{stem} * [N_{stem,min}]) & if\ N_{req,leaf} > NUP \end{cases} \tag{A13}$$



In equation (A13), the minimum function ensures that the stem N does not fall below its minimum. In the scenario that the leaves required more N than taken up by the crop, $N_{leaving,leaf} = 0$ and $N_{into,stem} = 0$.


**Table A3: Citations and names for equations describing N partitioning to the stem and leaf**

| Equation number | Equation name | Developed according to… |
|---|---|---|
| (A9) | The N required by the stem for growth | (Soltani & Sinclair, 2012) |
| (A10) | The N partitioned to the leaf and stem if the stem has a N deficiency and N uptake is not great enough to meet it | (Soltani & Sinclair, 2012) and adapted to remove leaf senescence releasing N in this study |
| (A11) | The N required for leaf area expansion | (Soltani & Sinclair, 2012) |
| (A12) | The N going into the leaf pool if N uptake met the stem N deficit | (Soltani & Sinclair, 2012) |
| (A13) | The N leaving the stem to maintain leaf area expansion under no stem N deficit | (Soltani & Sinclair, 2012) and adapted in this study to ensure stem N does not go below its minimum |




## A.3 N partitioning to the grain

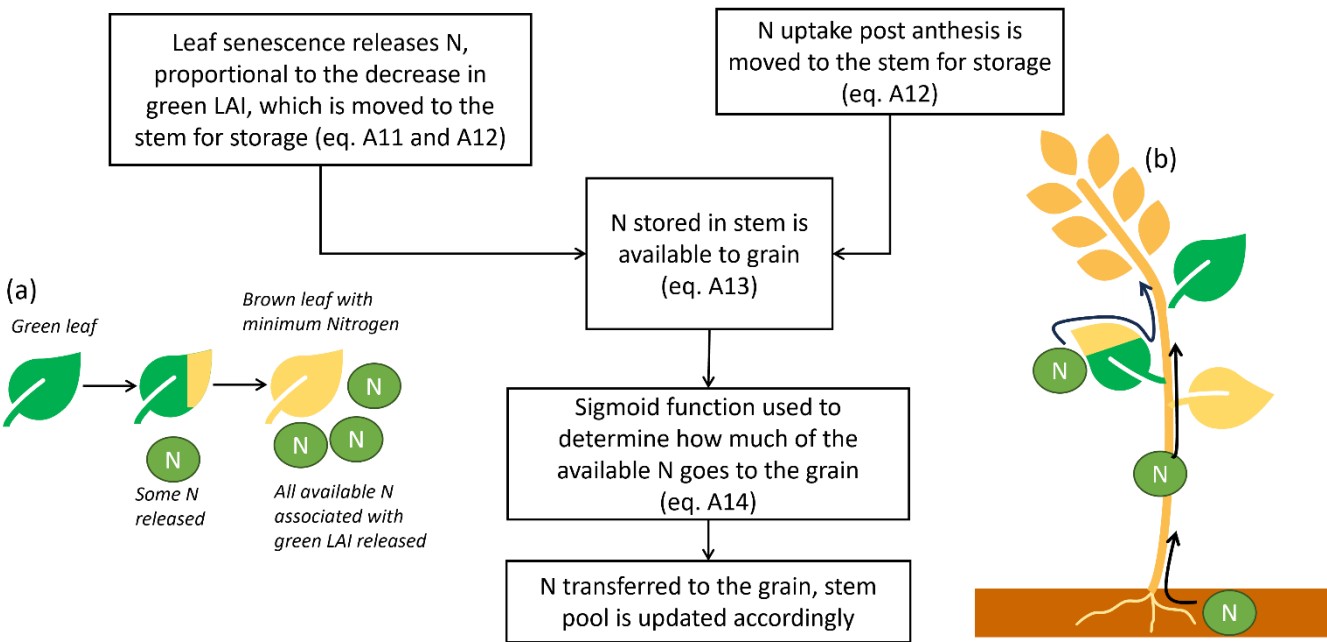

**Figure A2: Figure showing the allocation of N to the wheat grains post-anthesis with equations indicated appropriately. Sub-image (a) is a visual representation of a leaf releasing N as the leaf area senesces. Sub-image (b) shows the 3 locations where N is transferred to the grain from: post anthesis N uptake, N stored in the stem, senescing leaf area.**

After anthesis, the grain begins to fill with N and the model equations change to reflect that the priority is grain fill, not leaf area expansion or growth. After anthesis $N_{into,leaf} = 0$ always. As leaf area senesces, N is released:

$$N_{leaving,leaf} = \left(LAI_{yesterday} - LAI_{today}\right) * \left(\frac{N_{leaf}}{LAI_{yesterday}} - \left[N_{leaf,min}\right]\right) \tag{A14}$$

where $LAI_{yesterday}$ and $LAI_{today}$ represent the values of the leaf area index yesterday and today as calculated in DO₃SE-Crop. N released from the leaf, and N taken up by the plant post-anthesis are added to the stem N pool for storage, as they would have to travel through the stem to reach the grains (Sanchez-Bragado et al., 2017). The stem Nitrogen is then updated accordingly:

$$N_{into,stem} = N_{leaving,leaf} + NUP_{post} \tag{A15}$$

Not all of the N in the stem is available to be transferred to the grain as the stem has a minimum N concentration. Therefore, the available stem N (often referred to as the labile pool in other crop models) is calculated as:

$$N_{available} = N_{stem} - \left(DW_{stem} * \left[N_{stem,min}\right]\right) \tag{A16}$$

The fraction of $N_{available}$ that is transferred to the grain each day is determined through a sigmoid function. The sigmoid was chosen as it uses only two extra parameters which allows the start and rate of grain fill with N to be customised without the




addition of much complexity. The N in the $N_{available}$ pool can be transferred to the grain, or it can remain as part of the stem. The sigmoid determines the fraction of N going to the grain from $N_{available}$. The fraction increases as the plant develops.

Multiplying $N_{available}$ by the sigmoid function gives the amount of N transferred to the grain that day. $\alpha_N$ and $\beta_N$ are the coefficients that customise the onset and rate of grain fill. They can be calibrated to customise grain fill.

$$N_{to\_grain} = N_{available} * \frac{1}{1 + exp(-\alpha_N(dvi - \beta_N))} \tag{A17}$$

Once N has been transferred to the grain, the leaf and stem N pools are decreased according to their N availability to account for this transfer. **Error! Reference source not found.** diagrammatically represents the grain filling process and **Error! Reference source not found.** shows an illustration of the sigmoid function with varying parameterisations to control grain

fill.

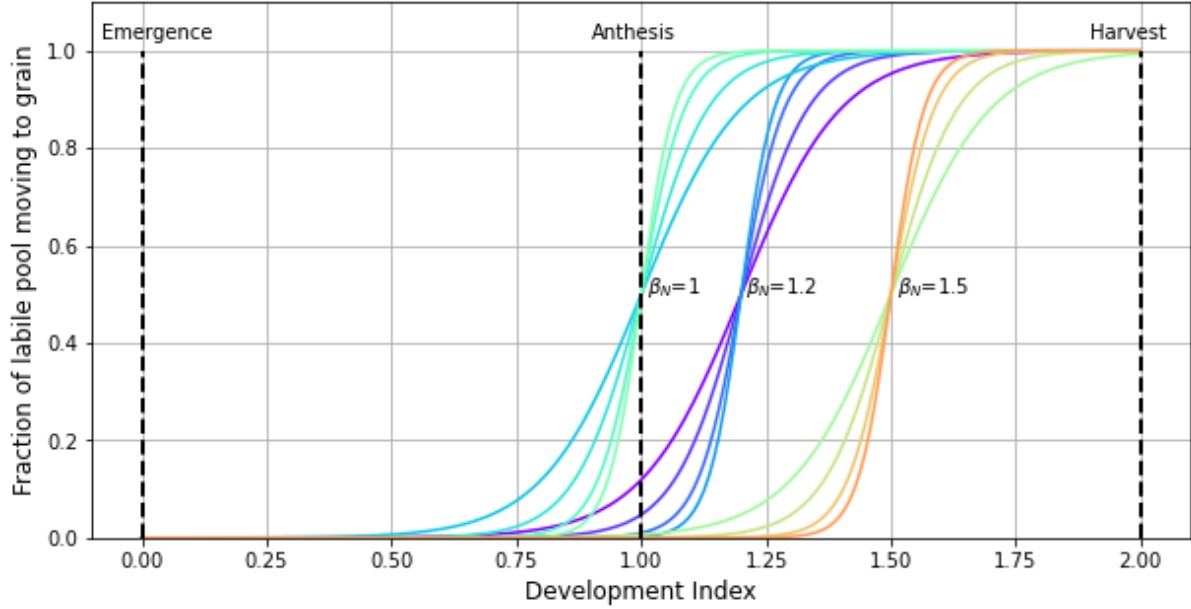

**Figure A3: Plot showing example parameterisations of $\alpha_N$ and $\beta_N$ for customising the sigmoid function describing the fraction of labile N moving to the grain. For each value of $\beta_N$, values of $\alpha_N$ of 10, 15, 23 and 30 are plotted to show how the rate, start and end of grain fill can be customised. Though $\beta_N = 1$ has been plotted to illustrate how the shape of the function can be customised, care**

**should be taken if using this value, as for lower values of $\alpha_N$ it can imply grain filling with N begins midway between emergence and anthesis. Sensible parameterisations should be chosen.**





**Table A4: Citations and names for equations describing grain filling with N**

| Equation number | Equation name | Developed according to… |
|---|---|---|
| (A14) | N released from senescing leaves | Based on the equations in Soltani & Sinclair (2012) describing release of N from the leaves, but adapted for this study to match desired variables |
| (A15) | Update stem N pools | (this study) |
| (A16) | Calculate available N in stem | Based on the equations in Soltani & Sinclair (2012)(Soltani & Sinclair, 2012) describing release of N from the stem, but adapted for this study to match desired variables and incorporate post anthesis N uptake |
| (A17) | Grain N sigmoid function | (this study) |

**Table A5: Parameterisation of alpha and beta parameters for grain filling**

| Parameter | Value used in this study | Source |
|---|---|---|
| $\alpha_N$ | 23 | (this study) |
| $\beta_N$ | 1.2 | (this study) |

**A.4 Ozone impact on N remobilisation**

The effect of ozone on grain N has been described in the main body of this study but will be discussed with relation to the

equations and how they link with the previously described model structure here. Broberg et al. (2017) and Brewster et al. (2024) found that as ozone concentrations increased, the fraction of N in the leaf and stem at harvest, that was present at anthesis, increased. In essence, the remobilisation of N from the leaves and stem to the grains decreased. A regression of the combined data from Broberg et al. (2017) and Brewster et al. (2024) representing the remobilisation is shown in Figure 2 of the main body of the study.

To represent the reduced remobilisation under increased ozone exposure, the remobilisation regression was used to calculate new values of $[N_{stem,min}]$ and $[N_{leaf,min}]$ under ozone exposure. The fraction of N remaining in the leaf and stem at harvest that was there at anthesis ($fN_{remob}$) is approximated by:

$$fN_{remob} = \frac{DM_{leaf,harv} * [N_{leaf,min}] + DM_{stem,harv} * [N_{stem,min}]}{DM_{leaf,anth} * [N_{leaf,targ}] + DM_{stem,anth} * [N_{stem,targ}]}$$    (A18)





Equation (A18) assumes the anthesis leaf and stem N concentration is the same as the target N concentration, as the first iteration of the model assumes no N limitation. Additionally, for calibration purposes, the "harvest" N concentration was
assumed to be the same as the minimum N concentration. In the model itself this would not occur as not all available N will be remobilised due to ozone effects on senescence. However, for calibration purposes equation (A18) is a good approximation. Using equation (A18), minimum values of leaf and stem N for the differing ozone treatments were manually altered for each ozone concentration, to estimate values for the N remobilisation (red points on Figure 2) that form a linear regression fitting inside the 95% CI. The leaf and stem minimum N concentrations of each red point were extracted. The minimum leaf N
concentration, and minimum stem N concentration, were regressed separately with M12 to give a regression describing how the minimum N concentration in the plant part varies with ozone concentration:

$$\frac{[N_{leaf,min}]}{1 \ gN \ LAI^{-1}} * 100 = m_{leaf} * \frac{[O_{3,M12}]}{1 \ ppb} + c_{leaf} \tag{A19}$$

$$\frac{[N_{stem,min}]}{1 \ gN \ DM^{-1}} * 100 = m_{leaf} * \frac{[O_{3,M12}]}{1 \ ppb} + c_{stem} \tag{A20}$$

**Table A6: Citations and names for equations describing the ozone effect on N remobilisation**

| Equation number | Equation name | Developed according to… |
|---|---|---|
| (A18) | The fraction of N remaining in the leaf and stem at harvest that was there at anthesis | (this study) |
| (A19) | Ozone effect on minimum leaf N concentration | (this study) |
| (A20) | Ozone effect on minimum stem N concentration | (this study) |

**Table A7: Parameterisation of parameters associated with the ozone effect on N grain filling**

| Parameter | Value | Source |
|---|---|---|
| $m_{leaf}$ | 0.798 | (this study) |
| $m_{stem}$ | 0.0138 | (this study) |
| $c_{leaf}$ | 10.89 | (this study) |
| $c_{stem}$ | 0.2293 | (this study) |



## A.5 Ozone impact on N remobilisation

In the DO$_3$SE-Crop model, green leaves are photosynthesising leaves, and brown leaves are senesced leaves. In the N module, the N associated with the senesced LAI is released (as described in section 1.3). Leaf area which has senesced will have the minimum leaf N concentration. Green leaf area which has not senesced will have a higher N concentration. An average leaf N

concentration can be calculated by taking the absolute N in both green and brown leaves and dividing it by the total (green + brown) leaf DM.

## Code availability

(citation to repository to be added before publication)

## Data availability

(citation to repository to be added before publication)

## Author contributions

Conceptualisation: Jo Cook, Lisa Emberson, Felicity Hayes, Samarthia Thankappan, Clare Brewster, Håkan Pleijel

Data curation: Jo Cook, Felicity Hayes, Clare Brewster

Formal analysis: Jo Cook

Methodology: Jo Cook, Lisa Emberson, Felicity Hayes, Samarthia Thankappan, Clare Brewster, Håkan Pleijel

Software: Jo Cook (N module and DO$_3$SE-Crop), Sam Bland (DO$_3$SE-Crop), Pritha Pande (DO$_3$SE-Crop), Nathan Booth (DO$_3$SE-Crop), Lisa Emberson (DO$_3$SE-Crop)

Supervision: Lisa Emberson, Felicity Hayes, Samarthia Thankappan

Visualisation: Jo Cook

Writing – original draft preparation: Jo Cook

Writing – review and editing: Jo Cook, Lisa Emberson, Felicity Hayes, Samarthia Thankappan, Clare Brewster, Håkan Pleijel, Nathan Booth, Pritha Pande, Sam Bland

## Funding

This work was supported by ACCE grant number NE/S00713X/1





**Competing interests**

The authors declare that they have no conflict of interest.

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
