# Peer review of "New ozone-nitrogen model shows early senescence onset is the primary cause of ozone-induced reduction in grain quality of wheat"

_EGUsphere, 2024_

## Referee Comment (RC1)

Review of

**New ozone-nitrogen model shows early senescence onset is the primary cause of ozone-induced reduction in grain quality of wheat**

by

Jo Cook et al.

The paper details the addition of nitrogen process into the DO3SE-Crop model. Nitrogen O3 interaction within a plant physiology is a very import element toward the definition of the yield of a crop and its quality.

The paper is complete and there are no major comments or suggestions to be made a part for the 3 listed below. It is supported by a thorough and complete literature, which well places the current study in terms of scientific relevance and novelty. The content is presented in a clear and structure way and it is supported by necessary appendixes.

This kind of modelling is complicated by definition, as it has to rely heavily on observation and parametrizations while aiming at a level of generality and applicability to a number of different cases. The authors have taken all necessary steps to make sure that they are in control of the parameter space and are able to identify rather precisely the relevant parameters that control the process.

In light of the above, the paper may be published though I would like 3 points to be clarified.

1- in the nice flow-chart representation of figure 1 the authors present the complete chain of causes and effects that relate O3 and N detailing what is in the model already, what is not and what has been added.

If I concentrate on box 1, I see for example that the process of neutralization of ozone uptake into the leaves is not modelled so my question is how do you calculate the excess of ROS not neutralised which then affects all other boxes (2 and 3)? How do you go from "Ozone enters the leaf" to "Accelerated senesce" in box 2 and "Reduced carboxyl oxidation" in box 3 in the actual model? It looks as if the chain that defines the storage of N is well represented in the new module but for ozone we go from entrance in the leaf to carboxyl efficiency reduction. Is the rest parameterised?

2- In the analysis of the Sensitivity results the authors mention:" . In our sensitivity analysis we observed a difference in the magnitude of S1 (the uncertainty in the output variable that is attributed to varying only that parameter) and ST (the uncertainty in the output variable that is attributed to varying a chosen parameter in combination with the other selected parameters) (Saltelli et al., 2008) between the different O3 treatments. It is unclear why this effect occurred. It isn't possible to determine whether the magnitude of S1 and ST is anomalous for the low or medium O3 treatments, or whether a pattern exists at all in S1 or ST between O3 treatments since the present study considers data on one cultivar for one location only." Can this discrepancy be attributed to a non-linear interaction between additional parameter added in the analysis and the other one? This would explain an increased sensitivity and could be determined by analysing the level of co-relation existing between the parameters used for the sensitivity. May be worth exploring.

3- This manuscript will be published as contribution to the TOAR Special Issue. However, I do not see a single reference to the project in the paper, which seems a bit odd in my humble opinion.  Clearly, it could be published as a standalone publication, though if the authors have chosen the TOAR SI they must have a reason and the readers should informed about it. Otherwise it looks like a "hopping on a freight train" (apologies for the analogy no offense intended) when you would have plenty of credit to afford a ticket for the first class wagon. I am sure there is a plausible explanation and a paragraph that links this beautiful work to TOAR should be added.

---

## Community Comment (CC1)

Comments by Owen R. Cooper (TOAR Scientific Coordinator of the Community Special Issue) on:

**New ozone-nitrogen model shows early senescence onset is the primary cause of ozone-induced reduction in grain quality of wheat**

Jo Cook, Clare Brewster, Felicity Hayes, Nathan Booth, Sam Bland, Pritha Pande, Samarthia Thankappan, Håkan Pleijel, and Lisa Emberson

EGUsphere [preprint], https://doi.org/10.5194/egusphere-2024-1311, 2024
Discussion started: 14 May 2024;  Discussion closes 26 July, 2024

This review is by Owen Cooper, TOAR Scientific Coordinator of the TOAR-II Community Special Issue. I, or a member of the TOAR-II Steering Committee, will post comments on all papers submitted to the TOAR-II Community Special Issue, which is an inter-journal special issue accommodating submissions to six Copernicus journals:  ACP (lead journal), AMT, GMD, ESSD, ASCMO and BG. The primary purpose of these reviews is to identify any discrepancies across the TOAR-II submissions, and to allow the author teams time to address the discrepancies.  Additional comments may be included with the reviews. While O. Cooper and members of the TOAR-II Steering Committee may post open comments on papers submitted to the TOAR-II Community Special Issue, they are not involved with the decision to accept or reject a paper for publication, which is entirely handled by the journal's editorial team.

**General Comments:**

TOAR-II has produced two guidance documents to help authors develop their manuscripts so that results can be consistently compared across the wide range of studies that will be written for the TOAR-II Community Special Issue.  Both guidance documents can be found on the TOAR-II webpage: https://igacproject.org/activities/TOAR/TOAR-II

*The TOAR-II Community Special Issue Guidelines*:   In the spirit of collaboration and to allow TOAR-II findings to be directly comparable across publications, the TOAR-II Steering Committee has issued this set of guidelines regarding style, units, plotting scales, regional and tropospheric column comparisons, tropopause definitions and best statistical practices.

*Guidance note on best statistical practices for TOAR analyses*:  The aim of this guidance note is to provide recommendations on best statistical practices and to ensure consistent communication of statistical analysis and associated uncertainty across TOAR publications. The scope includes approaches for reporting trends, a discussion of strengths and weaknesses of commonly used techniques, and calibrated language for the communication of uncertainty. Table 3 of the TOAR-II statistical guidelines provides calibrated language for describing trends and uncertainty, similar to the approach of IPCC, which allows trends to be discussed without having to use the problematic expression, "statistically significant".

**Specific Comments:**

The focus of this paper is outside my area of expertise and I do not have any specific comments regarding the details of the methodology or analysis. In terms of comparing the findings of this paper to the findings from TOAR-I, and to the other papers submitted to the TOAR-II Community Special Issue, I did not find any discrepancies.  I have listed a few detailed comments below to help with the Introduction.

Line 23
For non-experts, please spell out the full name of FAO (Food and Agriculture Organization of the United Nations)

Line 27
Regions of the world with high ozone concentrations relevant to wheat were identified by the first phase of TOAR (Mills et al., 2018)

Line 34
Fowler et al. (2008) is a very good paper, but it's now a little dated.  Current projections of surface ozone evolution are shown in Figure 6.20 in Chapter 6, IPCC AR6 WG-I, The Physical Science Basis (Szopa et al., 2021).  They show a lot of variability in surface ozone evolution depending on the emissions scenario. Under the SSP3-7.0 scenario (approximately business as usual) annual average surface ozone continues to increase in most regions (although this figure does not show projections for strong ozone pollution episodes).  Another consideration is presented by Zanis et al. (2022), who suggest that climate change could impose a "climate penalty" on surface ozone, with more frequent heatwaves exacerbating ozone in South and East Asia.

Lines 44-46
This sentence on grain protein is difficult to understand, please reword.

**References:**

Mills, G, et al. 2018. Tropospheric Ozone Assessment Report: Present-day tropospheric ozone distribution and trends relevant to vegetation. Elem Sci Anth, 6: 47. DOI: https://doi.org/10.1525/elementa.302

Szopa, S., V. Naik, B. Adhikary, P. Artaxo, T. Berntsen, W.D. Collins, S. Fuzzi, L. Gallardo, A. Kiendler-Scharr, Z. Klimont, H. Liao, N. Unger, and P. Zanis, 2021: Short-Lived Climate Forcers. In Climate Change 2021: The Physical Science Basis. Contribution of Working Group I to the Sixth Assessment Report of the Intergovernmental Panel on Climate Change [Masson-Delmotte, V., P. Zhai, A. Pirani, S.L. Connors, C. Péan, S. Berger, N. Caud, Y. Chen, L. Goldfarb, M.I. Gomis, M. Huang, K. Leitzell, E. Lonnoy, J.B.R. Matthews, T.K. Maycock, T. Waterfield, O. Yelekçi, R. Yu, and B. Zhou (eds.)]. Cambridge University Press, Cambridge, United Kingdom and New York, NY, USA, pp. 817–922, doi:10.1017/9781009157896.008

Zanis, P., Akritidis, D., Turnock, S., Naik, V., Szopa, S., Georgoulias, A.K., Bauer, S.E., Deushi, M., Horowitz, L.W., Keeble, J. and Le Sager, P., 2022. Climate change penalty and benefit on surface ozone: a global perspective based on CMIP6 earth system models. Environmental Research Letters, 17(2), p.024014.

---

## Author Comment (AC1)

We are grateful to the editor for their valuable comments and have modified the paper accordingly. We thank the editor for their comments which have improved the strength of the manuscript.

Line 23 For non-experts, please spell out the full name of FAO (Food and Agriculture Organization of the United Nations)

We have corrected this in the manuscript (Line 39 of the revised manuscript)

Line 27 Regions of the world with high ozone concentrations relevant to wheat were identified by the first phase of TOAR (Mills et al., 2018)

We have added this reference to strengthen the argument around high $O_3$ concentrations in key wheat growing seasons (Line 44 of the revised manuscript)

Line 34 Fowler et al. (2008) is a very good paper, but it's now a little dated. Current projections of surface ozone evolution are shown in Figure 6.20 in Chapter 6, IPCC AR6 WG-I, The Physical Science Basis (Szopa et al., 2021). They show a lot of variability in surface ozone evolution depending on the emissions scenario. Under the SSP3-7.0 scenario (approximately business as usual) annual average surface ozone continues to increase in most regions (although this figure does not show projections for strong ozone pollution episodes). Another consideration is presented by Zanis et al. (2022), who suggest that climate change could impose a "climate penalty" on surface ozone, with more frequent heatwaves exacerbating ozone in South and East Asia.

We have consulted the references and modified the sentence and citations accordingly to update the locations where $O_3$ concentrations will likely increase, and to incorporate the effect that climate change will have on $O_3$ production. To strengthen the argument we use the recommended sources above and also Fu and Tian (2019). (Lines 49-54 of the revised manuscript). We also refer to the first phase of TOAR to highlight the project

Fu, T. M. and Tian, H. (2019). Climate Change Penalty to Ozone Air Quality: Review of Current Understandings and Knowledge Gaps. *Current Pollution Reports*, 5 (3), pp.159–171. [Online]. Available at: doi:10.1007/s40726-019-00115-6.

Lines 44-46 This sentence on grain protein is difficult to understand, please reword

We agree with the editor that this sentence is confusing. We have broken it into 2 sentences and re-phrased it to enhance readability (Lines 60-63 of the revised manuscript)

---

## Author Comment (AC2)

We are grateful to the reviewer for their suggestions on our manuscript. We feel the manuscript is significantly improved as a result of their suggestions. We also thank them for their support and kind comments on the nature of the manuscript.

The paper details the addition of nitrogen process into the DO3SE-Crop model. Nitrogen O3 interaction within a plant physiology is a very import element toward the definition of the yield of a crop and its quality. The paper is complete and there are no major comments or suggestions to be made a part for the 3 listed below. It is supported by a thorough and complete literature, which well places the current study in terms of scientific relevance and novelty. The content is presented in a clear and structure way and it is supported by necessary appendixes. This kind of modelling is complicated by definition, as it has to rely heavily on observation and parametrizations while aiming at a level of generality and applicability to a number of different cases. The authors have taken all necessary steps to make sure that they are in control of the parameter space and are able to identify rather precisely the relevant parameters that control the process. In light of the above, the paper may be published though I would like 3 points to be clarified.

1 - in the nice flow-chart representation of figure 1 the authors present the complete chain of causes and effects that relate O3 and N detailing what is in the model already, what is not and what has been added. If I concentrate on box 1, I see for example that the process of neutralization of ozone uptake into the leaves is not modelled so my question is how do you calculate the excess of ROS not neutralised which then affects all other boxes (2 and 3)? How do you go from "Ozone enters the leaf" to "Accelerated senesce" in box 2 and "Reduced carboxyl oxidation" in box 3 in the actual model? It looks as if the chain that defines the storage of N is well represented in the new module but for ozone we go from entrance in the leaf to carboxyl efficiency reduction. Is the rest parameterised?

This is a really valuable point raised by the reviewer. While the antioxidant processes are not explicitly modelled, we do simulate short term effects of $O_3$ on reducing carboxylation efficiency and subsequently photosynthesis. For the short-term damage the plant has the capacity to recover overnight from this damage, depending on leaf age. The older the leaf is, the less its recovery capacity. We also simulate a long-term effect of $O_3$ on accelerating senescence. These processes are given in detail in another paper of TOAR2 but are generally modelled by considering a modifying factor that varies between 0 and 1 depending on accumulated $O_3$, leaf age and cultivar specific parameters for the short-term, long-term and recovery factors. For example, $O_3$ tolerant and sensitive varieties will have different parameterisations. We appreciate that it is not presently clear from the diagram. We have added a few sentences to the section describing the diagram which provide an overview of the $DO_3SE$-Crop $O_3$ damage processes and refer the reader to the other study in TOAR2 which describes the $DO_3SE$-Crop model. (Lines 117-121 of the revised manuscript)

2- In the analysis of the Sensitivity results the authors mention:" . *In our sensitivity analysis we observed a difference in the magnitude of S1 (the uncertainty in the output variable that is attributed to varying only that parameter) and ST (the uncertainty in the output variable that is attributed to varying a chosen parameter in combination with the other selected parameters) (Saltelli et al., 2008) between the different O3 treatments. It is unclear why this effect occurred. It isn't possible to determine whether the magnitude of S1 and ST is anomalous for the low or medium O3 treatments, or whether a pattern exists at all in S1 or ST between O3 treatments since the present study considers data on one cultivar for one location only."* Can this discrepancy be attributed to a non-linear interaction between additional parameter added in

the analysis and the other one? This would explain an increased sensitivity and could be determined by analysing the level of co-relation existing between the parameters used for the sensitivity. May be worth exploring.

We agree with the reviewer that there is certainly a non-linear interaction of some form, which given the complex and inter-connecting nature of crop modelling is unsurprising. The difficulty comes with ascertaining whether the shape of the non-linear response of the sensitivity indices to $O_3$ is typical. In the present study we parameterised the model for one growing location and one cultivar so we cannot conclude whether the response is typical. If future work reveals that this is a common response for several locations and wheat growing varieties then it would be interesting to investigate the crop modelling processes underlying this to explain the response. However, we feel this is beyond the scope of this particular study. We have modified the sentences describing the effect to more clearly explain this point. We also believe that the original text implies an expectation of a linear response and that the newly modified text will more clearly explain the nuance of the response between $O_3$ treatments (lines 587-591 of the revised manuscript).

3- This manuscript will be published as contribution to the TOAR Special Issue. However, I do not see a single reference to the project in the paper, which seems a bit odd in my humble opinion. Clearly, it could be published as a standalone publication, though if the authors have chosen the TOAR SI they must have a reason and the readers should informed about it. Otherwise it looks like a "hopping on a freight train" (apologies for the analogy no offense intended) when you would have plenty of credit to afford a ticket for the first class wagon. I am sure there is a plausible explanation and a paragraph that links this beautiful work to TOAR should be added.

We appreciate the reviewer's comments regarding the inclusion of the paper in TOAR2. We now begin the introduction (Lines 29-37) with reference to the previous phase of TOAR and the goals of the second phase. We reference TOAR2's goal of investigating the impacts of tropospheric ozone on human health and vegetation, which this paper addresses by its development of new methodologies to consider the interaction between $O_3$ and N processes in wheat, which will subsequently impact crop quality and human health. We hope this addition makes the link between the TOAR2 project and the present work clearer.

---

## Author Comment (AC3)

We are grateful to the reviewer for their suggestions which have substantially improved this manuscript. We thank the reviewer particularly for their comments which have improved the clarity and readability of the manuscript.

Cook and others incorporate a new model meant to explicitly simulate the impact of ozone on plant nitrogen dynamics and apply it to wheat trials. The results are interesting with a nice and defensible finding that stay green varieties are likely to minimize ozone impacts, which tend to hasten senescence. The manuscript could benefit from a number of adjustments that would make it easier to read and more concise.

The abstract was rather terse and did not enumerate particular findings, quantitatively, that make the study unique.

We thank the reviewer for bringing this to our attention. We have modified the abstract to incorporate the novelty of the study as well as highlighting the key findings (Lines 13-15, 16-18, 20-21)

Please edit the manuscript for flow and redundancies: note for example 'Northern India' is used twice in 28-29.

We have identified and removed several redundancies throughout the manuscript:

- Northern India twice (formerly Lines 28-29 now lines 45-46)
- "Figure 1 provides an overview of which processes are included already in the DO3SECrop model, which processes will be included in the new N module, and which processes will be excluded" (formerly Lines 98-99) -> "Figure 1 provides an overview of processes already included in $DO_3SE$-Crop, those to be added in the new N module, and those which are excluded" (lines 116-117 of the revised manuscript)
- "As a result of accelerated senescence leading to diminished photosynthesis, less photosynthate is produced (Emberson et al., 2018)" -> "Diminished photosynthesis leads to lesser photosynthate production" (line 138 of the revised manuscript)
- "Wheat yields are reduced due to the reduced photosynthesis and reduced duration of grain filling" (formerly Line 134) -> "Wheat yields decrease due to reduced photosynthesis and grain filling duration" (line 151 of the revised manuscript)

The narrative is tied to the FAO's sustainable development goals, but ozone impacts to wheat is important regardless and becomes a bit of a distraction because the purpose of the manuscript stands alone: the importance of ozone to wheat yield and productivity.

We have modified the sentence (formerly lines 34-35) to remove reference to sustainable development goals but retain the emphasis on $O_3$ impacts on wheat yields and quality

Note usage errors like the rogue period on line 52.

We have identified the rogue and corrected the punctuation

67: 'possess the capacity to' -> 'can'. Emphasis on removing all unnecessary words and phrases will make the manuscript more succinct and impactful.

We thank the reviewer for bringing this to our attention, we have removed several unnecessary words and phrases to improve clarity of the manuscript:

- "culminate in reductions to" (formerly Line 41) -> "reduce" (line 60 of the revised manuscript)
- "the reduction in yield that occurs under stressors" (formerly line 51) -> "yield reductions under stressors" (line 70 of the revised manuscript)
- "remobilisation of proteins" (formerly line 52) -> "protein remobilisation" (line 71 of the revised manuscript)
- "It is important to understand the mechanisms" (formerly Line 53) -> "Understanding the mechanisms.... is crucial" (line 71 of the revised manuscript)
- "A drawback of experimental work is that it is time consuming ..." (formerly Lines 57-58) -> "However, experimental work is time consuming..." (line 76 of the revised manuscript)
- "using fewer resources than would be required for experimental investigation" (formerly lines 60-61) -> "using fewer resources than required" (line 78 of the revised manuscript)
- "conversion factor, such as that from Mariotti, Tomé and Mirand (2008), to convert" (formerly Line 64) -> "conversion factor (e.g. Mariotti, Tome and Mirand (2008)) to convert" (line 82 of revised manuscript)
- "possess the capacity to simulate" (formerly line 67) -> "can simulate" (line 85 of the revised manuscript)
- "which could, in principle, be used to" (formerly Line 71) -> "which could be used to" (line 85 of the revised manuscript)
- "No model currently exists that includes the capacity to simulate the reduced remobilisation" (formerly Line 74) -> "Currently, no model simulates" (line 88 of the revised manuscript)
- "aid with" (formerly Line 97) -> "guide" (line 115 of the revised manuscript)
- "the response of the stomata" (formerly Line 104) -> "stomatal response" (line 125 of the revised manuscript)
- "which reduces" (formerly Line 104) -> "reducing" (line 125 of the revised manuscript)
- "causing damage to the" (formerly Line 106) -> "damaging" (line 126 of the revised manuscript)
- "The degradation of photosynthetic pigments by ROS" (formerly Line 108) -> "ROS degradation of photosynthetic pigments" (line 129 of revised manuscript)
- "The degradation of Rubisco by ROS" (formerly line 112) -> "ROS degradation of Rubisco" (line 133 of revised manuscript)
- "Accelerated senescence as a result of $O_3$ exposure can reduce the green leaf area available for photosynthetic reactions" (formerly Line 120) -> "$O_3$ induced accelerated senescence reduces the green leaf area for photosynthesis" (line 137 of the revised manuscript)
- "A larger proportion" (formerly Line 118) -> "more" (line 138 of the revised manuscript)
- "allocation of assimilate to flowers and seeds is prioritised in annual crops such as wheat" (formerly Line120) -> "annual crops, such as wheat, prioritise allocation of assimilates to flowers and seeds" (line 140 of the revised manuscript)
- "Generally, grain protein concentrations are increased under elevated $O_3$, resulting from a relatively smaller decrease in uptake and re-translocation of N relative to the $O_3$ induced decrease in grain dry matter" (formerly Lines 139-140) -> "Grain protein concentrations increase under elevated $O_3$, due to a smaller decrease in N uptake and re-translocation relative to the $O_3$-induced decrease in grain DM" (lines 158-159 of the revised manuscript)
- "to provide defence" (formerly Line 144) -> "defend" (line 163 of the revised manuscript)

- Approximately 5-10 days after mid-anthesis we see a reduction in grain DM accumulation for the higher $O_3$ concentrations (formerly Line 428-429) -> "Under higher $O_3$ concentrations, grain DM is reduced ~5-10 days after mid-anthesis." (lines 449-450 of the revised manuscript)
- "begins to be distinguishable" (formerly line 482) -> "are distinguishable" (line 494 of the revised manuscript)
- "Initially, there is a very rapid increase in grain N%" (formerly line 498) -> "Initially, grain N% increases rapidly (line 507 of revised manuscript)

Eq 1 and elsewhere: don't use the star in formal mathematical equations for multiplication, it has too many meanings (https://en.wikipedia.org/wiki/Asterisk#Mathematics)

We thank the reviewer for bringing this to our attention. We have now removed all use of "*" in equations and replaced them with the multiplication sign "x"

Table 1: just note in the legend that the parameters are unitless. There are also probably too many significant digits given realistic uncertainties.

We have removed the unit column of the table and stated in the legend that the parameters are unitless. We take the comment about parameter uncertainty. However, we cannot reduce the significant figures too much given the models sensitivity to these parameters. We now report the values to either 1 decimal place or 2 significant figures taking into consideration the sensitivity of the model to these parameters and the reviewers comment regarding uncertainties

What do the lightning bolts mean in figure 3?

We thank the reviewer for bringing this to our attention. We have modified the figure caption to explain that the lightning bolts represent the locations where $O_3$ affects N processes in the newly developed $DO_3SE$-CropN model (lines 264-265 of the revised manuscript)

320: 'that they were varied' -> 'between which they were varied'?

We have now modified the table caption to correctly explain this point (line 338 of the revised manuscript)

In Figure 4 and elsewhere, why do the 'simulated' variables have no uncertainty estimates?

Uncertainties in crop modelling are difficult to quantify and result from several sources. There are uncertainties associated with the input meteorological and $O_3$ concentration data used to run the model, associated uncertainties of the experimental data used for model parameterisation leading to uncertainty on calibrated model parameters. There will also be uncertainty due to the data availability and assumptions made during the calibration process. Finally, there will be uncertainty associated with the modelling processes themselves as it is not possible to perfectly replicate crop growth using a model (Chapagain et al., 2022). The most common method for identifying uncertainties is the sensitivity analysis (Saltelli et al., 2008), and the most commonly considered uncertainty source is the input data used (Chapagain et al., 2022). A sensitivity analysis identifying the variability in crop modelling outputs for $DO_3SE$-Crop that is attributed to differences in crop modelling inputs is underway and will also be published in TOAR2. As sensitivity and uncertainty analyses are complex, they are beyond the scope of the present study.

Chapagain, R. et al. (2022). Decomposing crop model uncertainty: A systematic review. Field Crops Research, 279 (June 2021), p.108448. [Online]. Available at: doi:10.1016/j.fcr.2022.108448.

Saltelli, A. et al. (2008). Global Sensitivity Analysis: The Primer. Chichester: John Wiley & Sons Ltd. [Online]. Available at: doi:10.1111/j.1751-5823.2008.00062_17.x.

345 and elsewhere: a time series figure of yield for different years could help the reader understand some of the variability involved.

We agree with the reviewers comment and have added an extra figure into the supplementary materials (Figure S5) which we refer the reader to in the text to aid with explanations of the variability in grain DM between years (lines 377-378 of the revised manuscript)

The Discussion makes some interesting points and is nice and upfront about the things that the model still struggles with. It could benefit from a bit more brevity if possible.

Based on prior remarks about redundancies and clarity, modifications to the discussion have been made (see response to prior comments). Further, we identified key areas in the discussion that could be summarised more clearly. The main sections that were edited are described below:

- Lines 432-445 in the original manuscript relating to drought stress effects on wheat DM accumulation profiles were summarised from 260 words to 163 words (lines 453-460 of the revised manuscript)
- Lines 461-473 in the original manuscript relating to simulations of grain N% were made more concise, reducing the word count from 271 to 218 (lines 475-485 of the revised manuscript)
- Lines 482-496 relating to profiles of grain N under O3 exposure were made more concise, reducing the word count from 286 to 229 (lines 494-510 of the revised manuscript)

I like the schematics, also in the appendix, that highlight what equations correspond to different processes.

We thank the reviewer for their kind comments regarding the schematics for the present work. We are pleased that they help convey understanding for the model.